# Biosynthesis of barley wax β-diketones: a type-III polyketide synthase condensing two fatty acyl units

Yulin Sun[1,4], Alberto Ruiz Orduna[2,4], Zhonghang Zhang[1], Sarah J. Feakins [3] & Reinhard Jetter [1,2] ✉

The surface coatings of cereal plants are dominated by waxy β-diketones crucial for drought resistance and, therefore, grain yield. Here, barley (*Hordeum vulgare*) wax analyses reveal β-diketone and associated 2-alkanol ester profiles suggesting a common $C_{16}$ 3-ketoacid precursor. Isotope analysis further shows that the major ($C_{31}$) diketone is synthesized from two plastidial $C_{16}$ acyl units. Previous studies identified a gene cluster encoding enzymes responsible for β-diketone formation in barley, but left their biochemical functions unknown. Various assays now characterize one of these enzymes as a thioesterase producing long-chain (mainly $C_{16}$) 3-ketoacids, and another one as a polyketide synthase (PKS) condensing the 3-ketoacids with long-chain (mainly $C_{16}$) acyl-CoAs into β-diketones. The two enzymes are localized to the plastids and Endoplasmic Reticulum (ER), respectively, implying substrate transfer between these two sub-cellular compartments. Overall, our findings define a two-step pathway involving an unprecedented PKS reaction leading directly to the β-diketone products.

Plants must seal the vast surfaces of their above-ground organs against water loss and pathogens[1]. They achieve this crucial eco-physiological function with waxes formed in their epidermis and deposited as a thin layer on the surface[1,2]. The waxes are complex mixtures of aliphatic compounds mostly derived from fatty acid metabolism, with aliphatic chains typically containing 24–34 carbons[3]. Most wax compounds have oxygen functional groups, predominantly at one chain end but also on mid-chain carbons. The composition of wax mixtures varies substantially between plant species and organs, leading to characteristic distributions of compound classes and chain lengths within each of them[3,4]. The common wax compounds are synthesized in three stages, involving (i) formation of $C_{16}$ fatty acyls by plastidial fatty acid synthases (FASs); (ii) further elongation to very-long-chain (VLC) acyl-CoAs by fatty acid elongases (FAEs) in the endoplasmic reticulum (ER); (iii) modification of the carboxyl headgroups on pathways involving either reduction to alcohols or decarbonylation to alkanes[2].

Many plant species accumulate specialty compounds in their surface waxes, frequently in the form of mid-chain β-diketones. Characteristic VLC β-diketones are found in diverse dicots (e.g., Myrtaceae, Buxaceae, Ericaceae, Caryophyllaceae, and Asteraceae)[5–10] and monocots (e.g., *Hosta* and *Vanilla* spp.)[11,12]. The wax β-diketones are especially widespread among the Poaceae[10], including crops such as wheat (*Triticum durum*, *T. aestivum*)[13–15], barley (*Hordeum vulgare*)[8,16], and rye (*Secale cereale*)[17]. For example, barley waxes contain predominantly $C_{31}$ 14,16-diketone, together with minor amounts of $C_{33}$ and $C_{29}$ diketones[8]. In many species, β-diketones are accompanied by esters of 2-alkanols thought to be side products of β-diketone formation (Fig. 1A)[10,18].

β-Diketones are produced in late vegetative and reproductive growth stages in Poaceae crops, especially in the waxes covering upper leaf sheaths, flag leaves and spikes[15,16,19,20]. There, they are among the most dynamically regulated wax components during drought conditions[21], associated with high water use efficiency and light

[1]Department of Botany, University of British Columbia, Vancouver, BC V6T 1Z4, Canada. [2]Department of Chemistry, University of British Columbia, Vancouver, BC V6T 1Z1, Canada. [3]Department of Earth Sciences, University of Southern California, 3651 Trousdale Pkwy, Los Angeles, CA 90089, USA. [4]These authors contributed equally: Yulin Sun, Alberto Ruiz Orduna. ✉e-mail: reinhard.jetter@ubc.ca

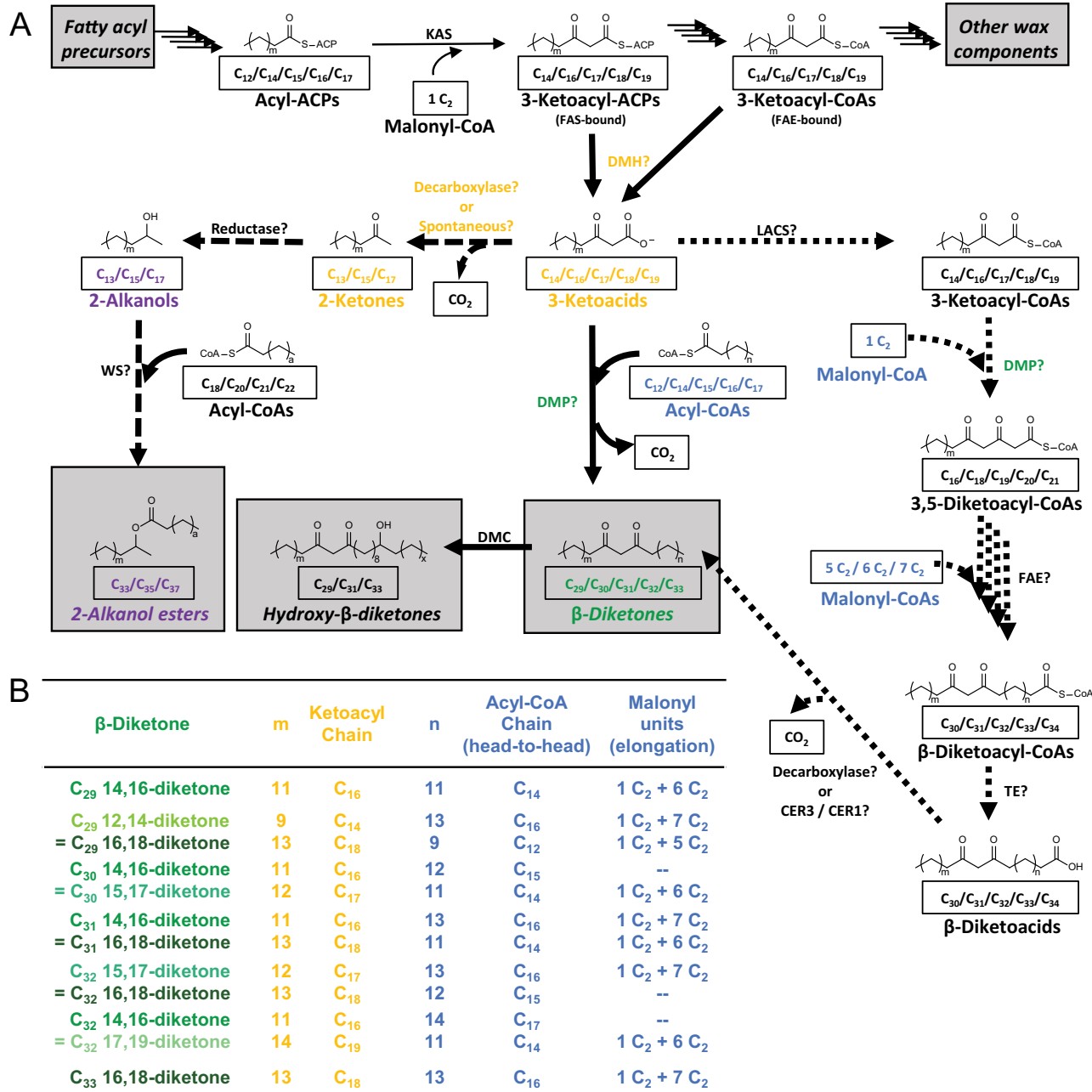

**Fig. 1 | Proposed β-diketone-forming pathways in barley. A** Pathways leading from acyl precursors (top left gray box) to various wax components (other gray boxes). Common wax components are formed by acyl-Acyl Carrier Protein (ACP) and acyl-CoA elongation (top row). The branch pathways leading to β-diketones and associated 2-alkanol esters (bottom half of scheme) proceed via central 3-ketoacid intermediates: diketone metabolism hydrolase (DMH) intercepts 3-ketoacyl-ACPs of plastidial Fatty Acid Synthase (FAS) or 3-ketoacyl-CoA intermediates of ER-bound Fatty Acid Elongase (FAE). The 3-ketoacids are either converted into 2-alkanol esters (left side; long-dashed arrows) or into β-diketones (right side). Two reaction paths from 3-ketoacids to β-diketones are feasible: (i) The elongation hypothesis (short-dashed arrows) involves activation by a Long-Chain Acyl-CoA Synthetase (LACS), condensation with a $C_2$ unit (from malonyl-CoA) by diketone metabolism polyketide synthase (DMP), further elongations catalyzed by FAE(s), and head group removal either by a thioesterase (TE) and a decarboxylase or by CER3/CER1-like enzymes. (ii) Alternatively, the head-to-head condensation hypothesis (solid arrows) predicts that DMP condenses fatty acyl-CoA starters and 3-ketoacid extenders. A diketone metabolism cytochrome-P450 (DMC) enzyme then likely hydroxylates β-diketones to produce hydroxy-β-diketones. **B** List of β-diketones in barley waxes, with isomers color-coded by diketo-group position. For isomers with n ≠ m two alternative names counting from either terminus are given. The chain length of the ketoacyl generating each β-diketone is given in orange. The acyl-CoA chain length required for head-to-head condensation to each β-diketone and the number of malonyl extender units required for the alternative elongation pathway are given in blue. Dashes indicate β-diketone isomers which cannot be synthesized by elongation. KAS β-ketoacyl-ACP synthase, WS wax ester synthase, CER3 ECERIFERUM 3, CER1 ECERIFERUM 1.

reflectance[22]. Therefore, β-diketones are crucial for maintaining grain yield under stress, and their accumulation is a favorable agronomic trait that has been highly selected in crop breeding[23]. However, the mechanism underlying β-diketone formation is only partially understood.

Early biochemical attempts to address β-diketone synthesis in barley with isotopic tracing suggested 3-ketoacids as key intermediates en route to $C_{31}$ 14,16-diketone (Fig. 1A and Supplementary Fig. S1)[8,24–26], but lacked definitive evidence for this assumption. Recently, genetic studies identified metabolic gene clusters involved in

β-diketone formation in both wheat and barley, each defined by the presence of three key genes. These genes were predicted to encode a hydrolase/carboxylesterase (named diketone metabolism hydrolase, DMH), a type-III polyketide synthase (PKS) (named diketone metabolism PKS, DMP) and a cytochrome-P450 oxidase (named diketone metabolism CYP450, DMC)[19,20]. In the process, the barley orthologs were recognized as the *Cer-q*, *Cer-c* and *Cer-u* genes described in earlier genetic studies, respectively[20,27]. Earlier mutant analyses had shown that the barley Cer-q/HvDMH and Cer-c/HvDMP enzymes were both required for β-diketone formation, and more recent VIGS experiments confirmed the in planta function of the corresponding enzymes also in wheat[19,20]. Thus, all previous evidence taken together suggested that DMH catalyzes the first step of β-diketone formation leading to the 3-ketoacid intermediate, and preliminary analyses confirmed that barley DMH may indeed intercept fatty acyl elongation intermediates to form 3-ketoacids[19]. While this provided first evidence for the enzymatic reaction leading to the key intermediate, it did not provide information on the later steps in the pathway.

Two fundamentally different mechanisms for the DMP-catalyzed formation of the diketo group must be considered (Fig. 1A and Supplementary Fig. S1). On the one hand, DMP may catalyze the condensation of 3-ketoacyl-CoA starters and malonyl-CoA extenders[10], resembling the activities of most PKSs[28]. The resulting 3,5-diketoacyl-CoA would require further elongation, likely by FAE-type elongase(s), and head group modification, similar to the pathways leading to common wax compounds. The major β-diketone of barley and wheat waxes, $C_{31}$ 14,16-diketone, may thus be formed via $C_{16}$ 3-ketoacid or $C_{18}$ 3-ketoacid as central pathway intermediate. However, both the nature of the intermediate and the enzymes modifying it in later steps of the predicted pathway remained elusive, despite the characterization of numerous barley β-diketone-deficient mutants. On the other hand, DMP could condense the 3-ketoacid intermediates with common fatty acyl-CoAs to directly form β-diketone products. Such a head-to-head condensation of two pre-formed hydrocarbon chains had been proposed early on[29], but various wax formation pathways were since shown to involve incremental build-up of hydrocarbon chains[2]. This second hypothesis for β-diketone biosynthesis thus diverges substantially from previously known wax formation mechanisms. Furthermore, it also invokes DMP substrates very different from those of most plant PKSs, which prefer aromatic starters and malonyl extenders[28,30,31]. However, it had been noted that the wheat DMP protein shares sequence similarity with *Curcuma longa* curcumin synthase (CURS)[19], a PKS enzyme known to use an aromatic ketoacid extender for head-to-head condensation with a second aromatic acyl[32]. Here, we aimed to revisit both these fundamental models for hydrocarbon chain formation and to elucidate the overall biosynthetic pathway leading to β-diketones in barley.

## Results

To elucidate the β-diketone biosynthesis pathway, we aimed to provide both chemical evidence defining the major pathway intermediates and biochemical evidence characterizing the key enzymes involved.

### Homolog and isomer distributions of β-diketones and 2-alkanol esters indicate the major pathway intermediate

In a first set of experiments, we sought to identify all β-diketone homolog and isomer structures accumulating in barley cv. Morex spike wax. To this end, the wax mixture was separated by thin-layer chromatography (TLC), and the fraction containing the β-diketones ($R_f$ 0.56) was analyzed by gas chromatography-mass spectrometry (GC-MS). The β-diketones were identified by characteristic base peaks *m/z* 100[33], while molecular ions and [M-18]$^+$ ions enabled the assignment of overall chain lengths $C_{29}$ to $C_{33}$ (Supplementary Fig. S2). Positional isomers for each homolog were assigned based on prominent

fingerprint fragments, leading to the identification of $C_{29}$ 12,14- and 14,16-diketone, $C_{31}$ 14,16-diketone and $C_{33}$ 16,18-diketone structures reported before[8]. In addition, novel even-numbered homologs were identified as $C_{30}$ 14,16-diketone and a mixture of $C_{32}$ 14,16- and $C_{32}$ 15,17-diketones. Analysis of the GC retention times showed that the even-numbered β-diketone homologs had straight hydrocarbon backbones devoid of methyl branches (Supplementary Fig. S3), implying that one of their acyl moieties must be formed from an unbranched, odd-numbered starter of de novo fatty acid synthesis.

GC-MS quantification showed that the spike wax β-diketones comprised 96.4% of $C_{31}$ 14,16-diketone, along with 0.4% $C_{29}$ 12,14-diketone, 0.3% $C_{29}$ 14,16-diketone and 2.6% $C_{33}$ 16,18-diketone (Fig. 2A and Supplementary Fig. S1). The $C_{30}$ 14,16-diketone, $C_{32}$ 14,16-diketone, and $C_{32}$ 15,17-diketone made up 0.1%, 0.2%, and 0.1% of the fraction, respectively. The corresponding TLC fraction of barley flag leaf sheath wax had a very similar β-diketone composition (Supplementary Table S1). Overall, the barely β-diketones comprised predominantly $C_{16}$ and $C_{18}$ ketoacyl units (Fig. 1B, m = 11 or 13), confirming $C_{16}$ 3-ketoacid and $C_{18}$ 3-ketoacid as most likely pathway intermediates.

To further assess the chain length distributions of key intermediates for β-diketone formation, we analyzed the 2-alkanol esters known to be pathway side- products. In a TLC fraction of barley spike wax ($R_f$ 0.73) both 1-alkanol esters and a series of 2-alkanol esters were identified. Among the 2-alkanol esters, the $C_{33}$, $C_{35}$, $C_{36}$, and $C_{37}$ homologs accounted for 8.3%, 58.5%, 3.0%, and 30.3%, respectively (Fig. 2B and Supplementary Table S2). Each of the four ester homologs comprised $C_{15}$ 2-alkanol, amounting to 88%, 96%, 100%, and 59%, respectively (Fig. 2B). Across all homologs, the esters contained 84% $C_{15}$ 2-alkanol, along with only 1% $C_{13}$ and 14% $C_{17}$ 2-alkanols. Similar 2-alkanol ester homolog distributions were observed in barley flag leaf sheath wax (Supplementary Table S3). Based on the general understanding that 2-alkanols are derived from corresponding 3-ketoacids one carbon longer[19,24], the high concentration of $C_{15}$ 2-alkanol esters suggests a precursor pool dominated by $C_{16}$ 3-ketoacid rather than $C_{18}$ 3-ketoacid.

### Mutant wax and carbon isotope analyses test elongation of the 3-ketoacid intermediate

To test whether β-diketone formation requires ER-localized VLC acyl elongation, we analyzed the wax of flag leaf sheaths of the *emr1* mutant lacking the core enzyme of the FAE complex responsible for wax precursor elongation, β-ketoacyl-CoA synthase 6 (KCS6/CER6). The mutant wax mixture showed a distinct elongation phenotype, with chain length distributions shifted to shorter homologs in comparison with the corresponding wild type (*H. vulgare* cv. Ingrid). All compound classes comprising chain lengths longer than $C_{24}$ in the wild type (alkanes, 1-alkanols, 1-alkanol esters, and alkylguaiacols[34]) had profiles shifted towards shorter homologs in the mutant (Supplementary Fig. S4), similar to previous reports on the leaf wax of these lines[35]. However, the $C_{31}$ β-diketones and hydroxy-β-diketones dominating the flag leaf sheath wax were found in similar concentrations in both lines. Formation of their carbon backbones is thus independent of the major wax ER-elongation enzyme, and our findings do not support elongation of a ketoacid precursor towards β-diketones.

Plant lipids have characteristic $^{13}C$ depletion patterns known to encode information on growth conditions as well as biosynthetic mechanisms[36]. In particular, previous studies revealed differential isotope incorporation into wax compound classes and chain lengths in various biosynthesis steps[37–39]. Therefore, we aimed to analyze the isotope composition of β-diketones by GC coupled with isotope ratio mass spectrometry (IRMS), and to compare it with the isotope compositions of wax constituents formed along known biosynthesis pathways. The latter (alkanes and esters) had $\delta^{13}C$ values positively correlated with the amounts of carbon incorporated during plastidial fatty acid synthesis ($r^2 = 0.60$) (Fig. 2C). The β-diketones (>95% $C_{31}$

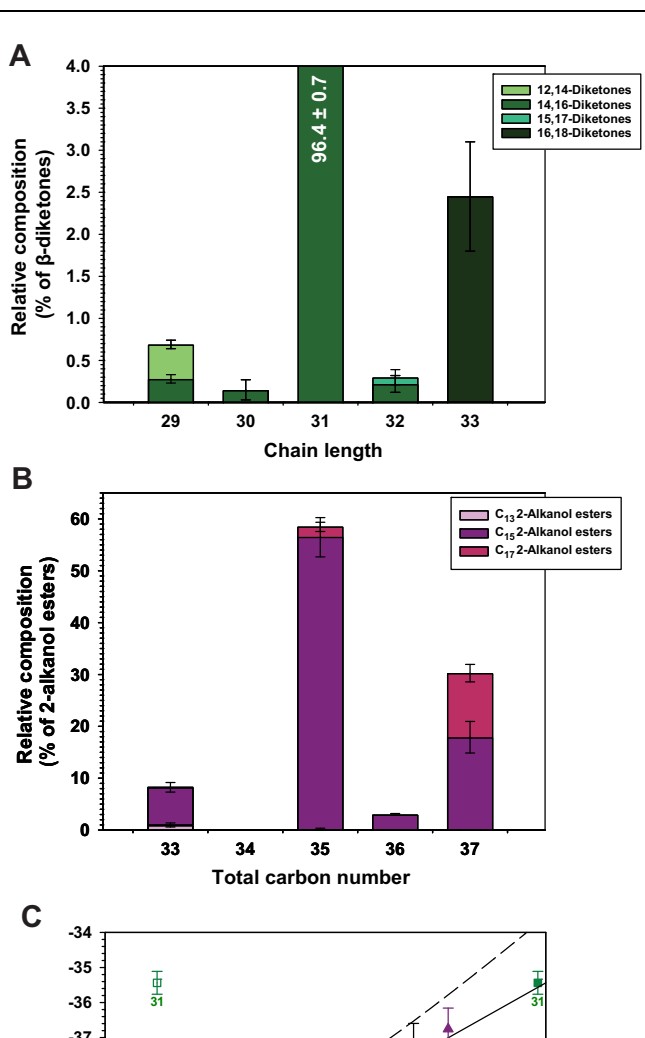

**Fig. 2 | Chemical characterization of the β-diketones and associated 2-alkanol esters in the wax mixture covering barley cv. Morex spikes. A** Relative amounts of β-diketone homologs (determined by GC-MS analysis of the characteristic fragment $m/z$ 100) and isomers (determined using characteristic α-fragments). **B** Relative amounts of 2-alkanol ester homologs and isomers. **C** Correlation between $^{13}C/^{12}C$ ratios and plastidial carbon incorporation of different wax compositions. $^{13}C/^{12}C$ ratios are reported in standard delta notation as $\delta^{13}C = (^{13}C/^{12}C$ sample $-^{13}C/^{12}C$ standard)/$^{13}C/^{12}C$ standard as permil (‰), where the standard is Vienna Pee Dee Belemnite. The $\delta^{13}C$ values of compounds (carbon numbers given next to each data point) with known biosynthesis pathways were plotted as a function of their plastidial carbon percentages, assuming elongation in the plastids up to $C_{16}$ and further elongation in the ER to the final chain lengths. Alkanes are formed by assembly of $C_{16}$ chains in plastids and further carbon addition in the ER[2], so the $C_{25}$, $C_{27}$, $C_{29}$, and $C_{31}$ alkanes contain 64%, 59.3%, 55.2%, and 51.6% plastidial carbon, respectively. Similarly, the 1-alkanol and 2-alkanol esters consist of alkyl and acyl moieties each incorporating a plastidial $C_{16}$ precursor, and % of plastidial carbon can be calculated accordingly. The Ordinary Least Squares regression for the alkanes and esters is shown as solid line ($r^2 = 0.60$), with 95% confidence intervals as dashed lines. Values for $C_{31}$ β-diketone are plotted for the elongation hypothesis (Fig. 1) assuming 51.6% plastidial carbon (left) or the head-to-head condensation hypothesis predicting 100% plastidial carbon (right). Data are presented as means ± standard deviations of three biological replicates for (**A**, **B**) and three analytical replicates for (**C**). Source data are provided as a Source Data file.

respective unsaturated $C_{15}$ as well as saturated $C_{13}$ and $C_{15}$ compounds (Fig. 3B, C). Further MS analysis of dimethyldisulfide (DMDS) adducts identified the unsaturated 2-ketones as isomers with double bonds mainly at the ω−7 position (Supplementary Fig. S6). The series of 3-hydroxyacids comprised predominantly saturated $C_{14}$ and $C_{16}$ as well as unsaturated $C_{16}$ compounds (Fig. 3D). None of the 2-ketones and 2-alkanols could be detected in corresponding empty-vector controls, while the concentrations of all 3-hydroxyacids were increased ~twofold from controls to *HvDMH*-expressing lines. As 2-ketones and 2-alkanols are likely formed by decarboxylation of respective 3-ketoacids and 3-hydroxyacids one carbon longer, our results imply that HvDMH generated predominantly $C_{16}$ 3-ketoacid and 3-hydroxyacid in *E. coli*. These HvDMH chain length preferences matched our previous results showing that the wax β-diketones and 2-alkanol esters originated mainly from $C_{16}$ ketoacyls.

To compare the chain length profiles of 3-keto-functionalized products with those of the available *E. coli* acyl pools, the latter were assessed by fatty acid methyl ester (FAME) analysis. The total lipids of the *HvDMH* expressors and empty-vector control lines had similar acyl profiles, with saturated $C_{12}$–$C_{18}$ fatty acyls peaking at $C_{16}$ and corresponding unsaturated acyls peaking at $C_{18}$ (Fig. 3E).

To corroborate the 3-ketoacyl thioesterase activity of HvDMH, we performed in vitro assays similar to those reported for characterization of tomato methylketone synthase[40,41]. In these coupled assays, purified *E. coli* acyl-carrier-protein (EcACP), *E. coli* malonyl-CoA:ACP transferase (EcFabD) and *Mycobacterium tuberculosis* 3-ketoacyl-ACP synthase (MtFabH) were used to convert malonyl-CoA substrate into malonyl-ACP and, together with $C_{14}$ acyl-CoA as second substrate, further into $C_{16}$ 3-ketoacyl-ACP. After incubation of this assay mixture with HvDMH, lipids were extracted and derivatized using elevated temperatures to decarboxylate the thioesterase product, sensitive 3-ketoacid, into corresponding 2-ketones for reliable quantification. In the assay product mixture, $C_{15}$ 2-ketone was identified based on GC retention time (Fig. 4A) and MS fragmentation patterns (Fig. 4B) identical to those of an authentic standard. Control in vitro assays with boiled HvDMH protein yielded small amounts of $C_{15}$ 2-ketone (Fig. 4C), while further control assays lacking MtFabH contained no detectable 2-ketone. Taken together, these assays thus defined a background of $C_{15}$ 2-ketone formed by hydrolysis of $C_{16}$ 3-ketoayl-ACP and decarboxylation of the resulting free 3-ketoacid, independent of HvDMH activity. Assays with the intact HvDMH enzyme contained significantly more $C_{15}$ 2-ketone than the boiled-enzyme controls, thus showing the

14,16-diketone) had $\delta^{13}C$ −35.3 ± 0.3‰, a value fitting the trend of all other compound classes only if assuming 100% plastidial carbon content in the β-diketone. Thus, our isotope analysis further corroborates that β-diketone formation does not involve elongation of 3-ketoacid intermediates by ER-residing FAE enzymes (c.f. Fig. 1 and Supplementary Fig. S1).

## Thioesterase activity and chain length preference of barley DMH (HvDMH)

To further establish the chain length distribution of the β-diketone synthesis intermediates, we analyzed the product profiles of the barley DMH enzyme (HvDMH) thought to catalyze the first reaction of the pathway. GC-MS analyses of *E. coli* expressing *HvDMH* identified saturated and monounsaturated $C_{13}$–$C_{17}$ 2-ketones and 2-alkanols, along with $C_{12}$–$C_{16}$ 3-hydroxy fatty acids (Fig. 3A and Supplementary Fig. S5). The 2-alkanol and 2-ketone series were dominated by

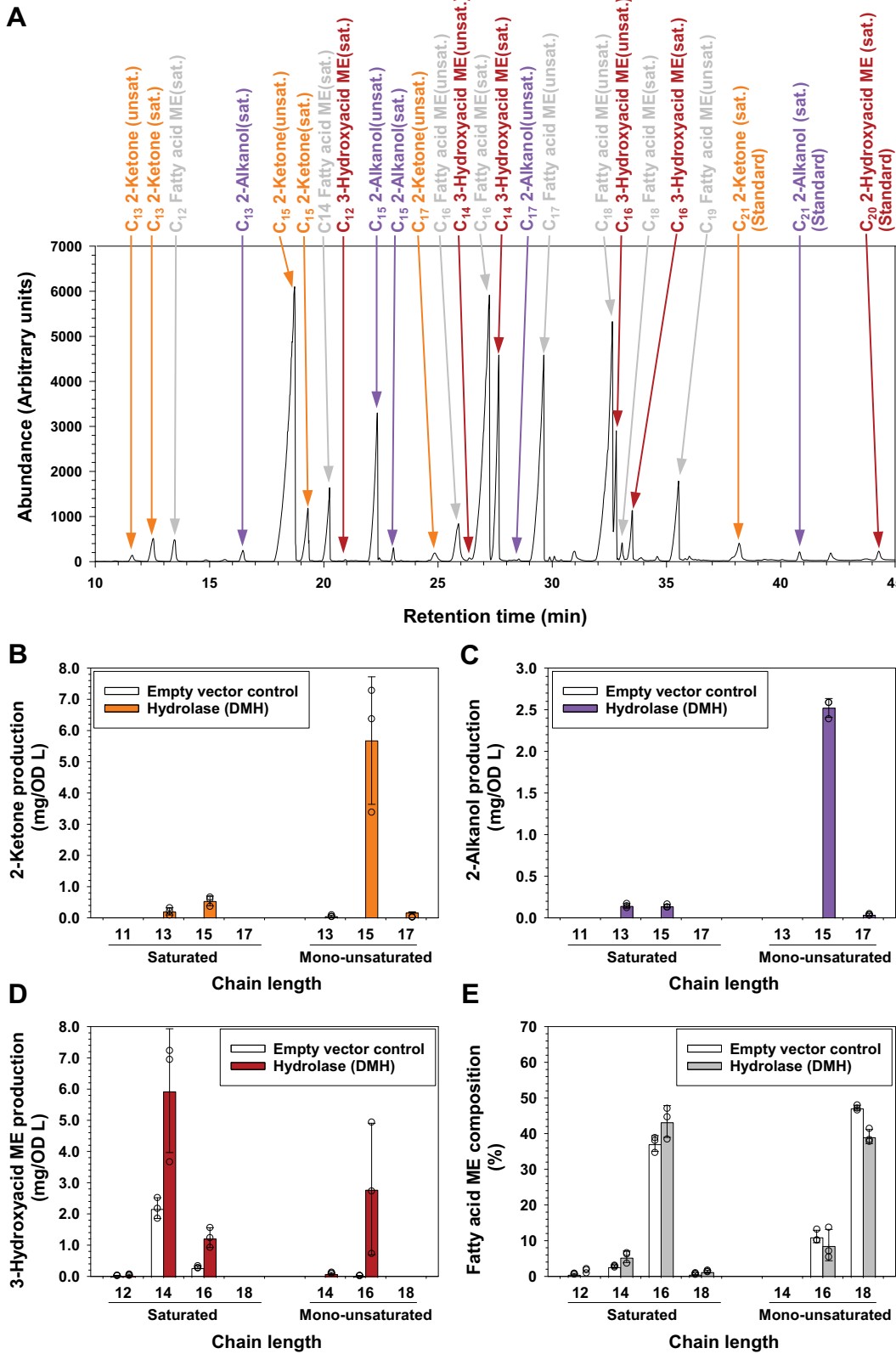

**Fig. 3 | Characterization of the barley diketone metabolism hydrolase (HvDMH) producing the key intermediates on the β-diketone synthesis pathway. A** Total ion chromatogram of lipids from *E. coli* expressing *HvDMH*, after transesterification into methyl esters (MEs) and conversion of hydroxyl groups into trimethylsilyl (TMS) ethers. Homolog series of saturated (sat.) and monounsaturated (unsat.) 2-ketones (orange), 2-alkanols (purple), 3-hydroxyacid MEs (red) and fatty acid MEs (gray) were detected. **B** Amounts of 2-ketones in *E. coli* expressing *HvDMH*

(quantified against internal standard $C_{21}$ 2-ketone). **C** Amounts of 2-alkanols in *E. coli* expressing *HvDMH* (quantified against internal standard $C_{21}$ 2-alkanol).
**D** Amounts of 3-hydroxyacid MEs in *E. coli* expressing *HvDMH* (quantified against internal standard $C_{20}$ 2-hydroxyacid ME and normalized to $C_{16}$ 3-hydroxyacid ME).
**E** Amounts of fatty acid MEs in *E. coli* expressing *HvDMH*. Data are presented as means ± standard deviations of three biological replicates. Source data are provided as a Source Data file.

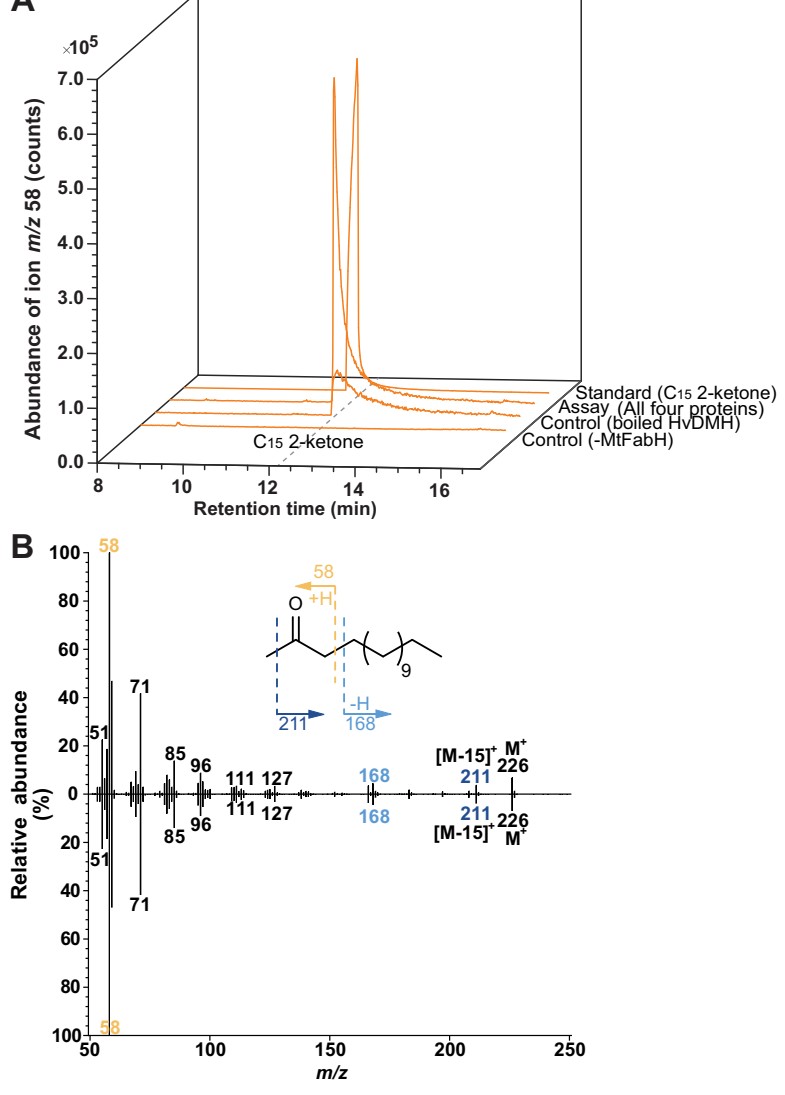

**Fig. 4 | In vitro characterization of barley diketone metabolism hydrolase (HvDMH). A** Gas chromatograms of the characteristic ion $m/z$ 58 of 2-ketones. Trace of the standard is compared with that of extracted assay product. Barley HvDMH enzyme was incubated with *E. coli* acyl carrier protein (EcACP), *E. coli* malonyl-CoA:ACP transferase (EcFabD), and *Mycobacterium tuberculosis* 3-ketoacyl-ACP synthase (MtFabH), and with $C_{14}$ fatty acyl-CoA and malonyl-CoA substrates. An identical assay with boiled HvDMH protein served as control for the hydrolase activity, and an assay lacking the MtFabH enzyme tested for background

levels of 3-ketoacyl-ACP and its decarboxylation product $C_{15}$ 2-ketone. **B** Mass spectra and fragmentation pattern of the $C_{15}$ 2-ketone peaks of the HvDMH assay in (**A**, upper panel) and standard (lower panel). **C** Quantification of $C_{15}$ 2-ketone in HvDMH assays shown in (**A**). Product amounts are normalized against the assay containing active HvDMH and MtFabH. Data are presented as means ± standard deviations of four biological replicates. Asterisks indicate significant differences between samples, as calculated by two-tailed Student's *t* test; *$P < 0.05$. Source data are provided as a Source Data file.

thioesterase activity of the enzyme (Fig. 4C). Overall, our assays demonstrated that HvDMH has in vitro thioesterase activity catalyzing the hydrolysis of 3-ketoacyl-ACP into free 3-ketoacid intermediates of β-diketone biosynthesis.

**Biochemical function of the barley DMP (HvDMP)**

Based on all the evidence suggesting $C_{16}$ 3-ketoacid as central intermediate of the β-diketone pathway, our next goal was to test its role as a substrate of the ensuing step and, thus, to establish the biochemical activity of HvDMP. However, the expression of HvDMH in yeast (*Saccharomyces cerevisiae*) did not lead to the production of the ketoacid, so the compound had to be supplemented exogenously. When $C_{16}$ 3-ketoacid was fed to yeast expressing *HvDMP*, three compounds were detected which were not present in controls expressing empty vector or lacking substrate. Based on their GC-MS characteristics, the major product was identified as $C_{31}$ 14,16-diketone, along with $C_{29}$ 14,16-

diketone and a monounsaturated $C_{31}$ 14,16-diketone (Fig. 5A, B). Thus, HvDMP proved active in yeast, likely condensing $C_{16}$ 3-ketoacid directly with $C_{16}$ fatty acyl to form $C_{31}$ 14,16-diketone.

To corroborate the biochemical activity of HvDMP, the recombinant enzyme was assayed in vitro with various substrate combinations. Incubation with $C_{16}$ ketoacid and $C_{16}$ fatty acyl-CoA gave a single product, identified as $C_{31}$ 14,16-diketone (Fig. 5B, C). In contrast, corresponding assays of HvDMP with varying molar ratios of $C_{16}$ ketoacid and malonyl-CoA yielded no detectable β-diketone or triketide/tetraketide products. Incubation of the enzyme with $C_{16}$ fatty acyl-CoA and malonyl-CoA in diverse concentrations also did not give detectable products. Together, these findings confirm that HvDMP uses a ketoacid instead of malonyl-CoA extenders for condensation with a fatty acyl-CoA.

To test the HvDMP-catalyzed condensation of the two substrates further, we fed per-deuterated $C_{16}$ fatty acid together with $C_{16}$ ketoacid to yeast expressing the enzyme. The resulting lipids contained

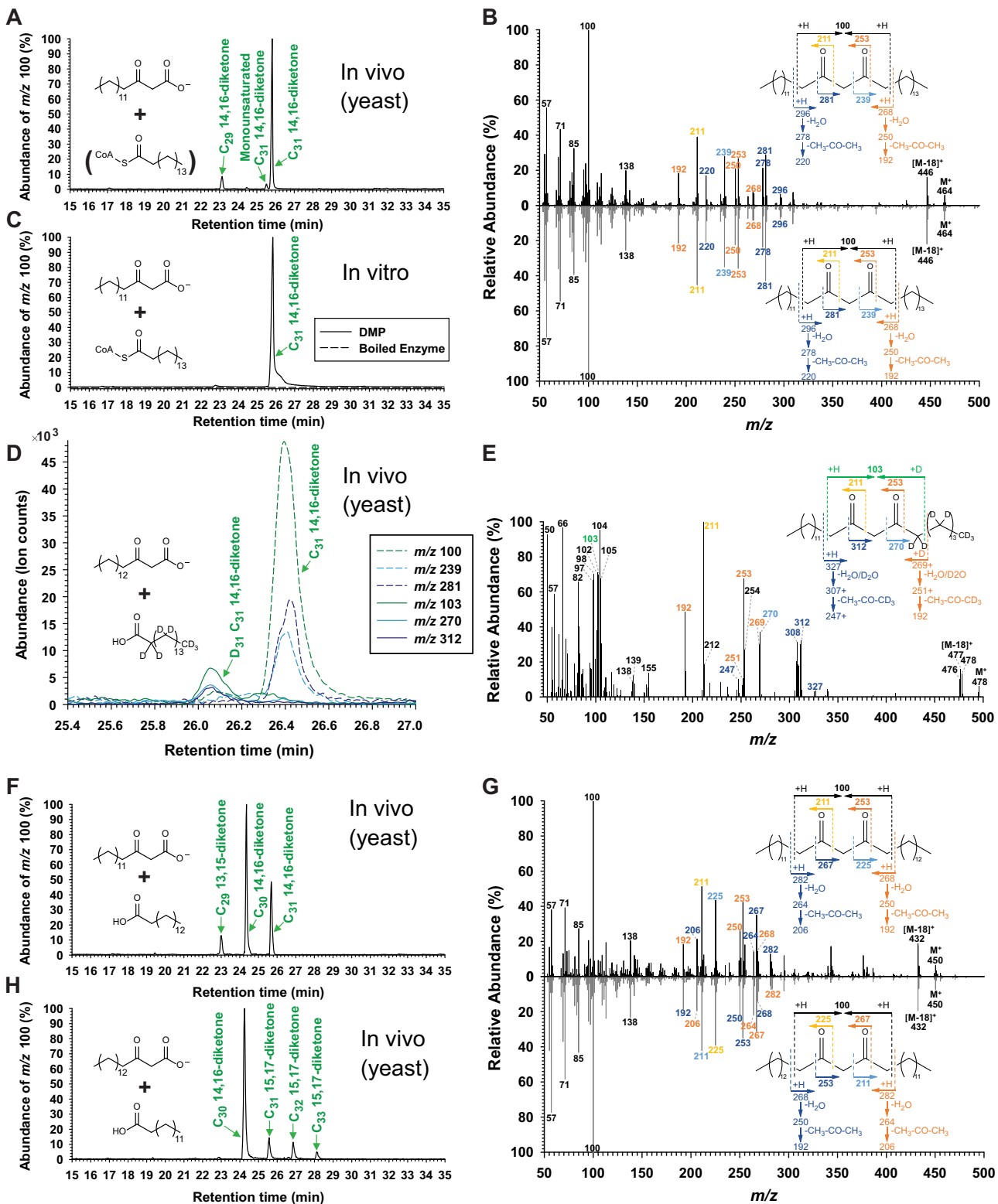

unlabeled $C_{31}$ 14,16-diketone together with a novel compound with shorter GC retention time (Fig. 5D). The MS characteristics (Fig. 5E and Supplementary Fig. S7) unambiguously identified this compound as $C_{31}$ 14,16-diketone with one normal ($C_{13}H_{27}$) and one per-deuterated hydrocarbon tail ($C_{15}D_{31}$) originating from the labeled fatty acid. The formation and exact structure of this product indicated the direct incorporation of the entire acyl moiety into the β-diketone product and further confirmed that HvDMP catalyzed its condensation with ketoacid also in vivo.

To test whether HvDMP condensed entire ketoacid and fatty acyl-CoA units, we assayed the enzyme with substrates having unusual odd-numbered acyl chains. In one experiment, feeding $C_{15}$ fatty acid and $C_{16}$ ketoacid to yeast expressing *HvDMP* yielded three compounds absent from empty-vector controls, the major compound being $C_{30}$ 14,16-diketone and the two minor ones $C_{29}$ 14,16-diketone and $C_{31}$ 14,16-diketone (Fig. 5F, G). In a similar assay, $C_{17}$ ketoacid and $C_{14}$ fatty acid were fed to *HvDMP*-expressing yeast, resulting in the formation of $C_{30}$ 14,16-diketone as a major product together with $C_{31}$ 15,17-diketone, $C_{32}$

**Fig. 5 | Characterization of the barley diketone metabolism polyketide synthase (HvDMP) catalyzing the condensation of 3-ketoacids and acyl-CoAs into β-diketones.** GC-MS analysis of in vivo and in vitro assays feeding acyl-CoA and/or ketoacid substrates to HvDMP. Chromatograms of fragment $m/z$ 100 selectively reporting β-diketones are shown on the left and mass spectra of key products on the right. **A** Chromatogram of lipids from yeast expressing *HvDMP* and supplemented with $C_{16}$ 3-ketoacid ($n = 11, 13$). **B** Mass spectra of $C_{31}$ 14,16-diketone peaks in (**A**, upper panel) and (**C**, lower panel). **C** Chromatogram of lipids from HvDMP in vitro assay with $C_{16}$ 3-ketoacid and $C_{16}$ acyl-CoA. Boiled recombinant enzyme incubated with $C_{16}$ 3-ketoacid and $C_{16}$ acyl-CoA served as control. **D** Chromatograms of characteristic fragments of lipids from yeast expressing *HvDMP* and supplemented with $C_{16}$ 3-ketoacid and per-deuterated fatty acid $C_{15}D_{31}COOH$, selectively reporting undeuterated (dashed lines) and deuterated β-diketones (solid lines). **E** Mass spectrum of the $D_{31}$-labeled $C_{31}$ 14,16-diketone in (**D**),
showing α-fragment, McLafferty rearrangement and water/acetone loss ions diagnostic for the $C_{14}$ acyl moiety of $C_{31}$ 14,16-diketone. Analogous fragments of the other hydrocarbon tail were shifted by 31 amu relative to the unlabeled $C_{31}$ 14,16-diketone, as were the molecular ion and the water loss ion [M-18]⁺. A cluster of ions around $m/z$ 103 is consistent with double-McLafferty rearrangement on both sides of the β-diketo group, a double-deuterated α-methylene and γ-deuterium transfer. Together, the GC and MS characteristics unambiguously identified this compound as $C_{31}$ 14,16-diketone with $C_{13}H_{27}$ and $C_{15}D_{31}$ hydrocarbon tails. **F** Chromatogram of lipids from yeast expressing *HvDMP* and supplemented with even-numbered 3-ketoacid ($C_{16}$) and odd-numbered fatty acid ($C_{15}$). **G** Mass spectra of $C_{30}$ 14,16-diketone in (**F**, upper panel) and (**H**, lower panel). **H** Chromatogram of lipids from yeast expressing *HvDMP* and supplemented with odd-numbered 3-ketoacid ($C_{17}$) and even-numbered fatty acid ($C_{14}$). Arabidopsis *LACS1* was expressed in all the yeast assays to enhance exogenous substrate uptake[61].

16,18-diketone and $C_{33}$ 15,17-diketone (Fig. 5G, H). The finding that each of these assays yielded several products sharing one acyl moiety, taken together with the structure of respective major β-diketone products, further confirmed that HvDMP catalyzed the direct condensation of fatty acyl-CoA and ketoacid substrates.

## Chain length preference of HvDMP for fatty acyl-CoA and 3-ketoacid substrates

Four experiments were performed to test the chain length preferences of HvDMP. In a first set of assays, we supplemented yeast expressing *HvDMP* with equal molar amounts of $C_{14}$, $C_{16}$, or $C_{18}$ ketoacids and quantified the resulting β-diketone products. Feeding of $C_{14}$ 3-ketoacid led to a β-diketone mixture comprising mainly $C_{27}$ 12,14-diketone, whereas $C_{16}$ 3-ketoacid feeding afforded mainly $C_{31}$ 14,16-diketone, and $C_{18}$ 3-ketoacid mainly $C_{33}$ 16,18-diketone (Fig. 6A, B). Thus, the HvDMP preferences for acyl-CoA substrates depended on the 3-ketoacid present, resulting in condensation of $C_{14}$ 3-ketoacid mainly with $C_{14}$ acyl-CoA, $C_{16}$ 3-ketoacid with $C_{16}$ acyl-CoA, and $C_{18}$ 3-ketoacid with $C_{14}$ and $C_{16}$ acyl-CoAs.

To gauge the effect of fatty acyl chain lengths available in the in vivo assays, we quantified the fatty acyls in the total lipid mixture of the yeast lines (as FAMEs). The acyl pools of control yeast (without substrate feeding) comprised mainly monounsaturated $C_{16}$ and $C_{18}$ chains (together 88%), along with saturated $C_{12}$–$C_{18}$ acyls dominated by $C_{16}$ (8%) and $C_{18}$ (3%) (Fig. 6C). Compared with this profile of available substrates, HvDMP greatly preferred saturated over unsaturated acyl substrates in our assays, and to some degree $C_{14}$ acyls over the more abundant $C_{16}$ and $C_{18}$ homologs (Supplementary Fig. S8A).

The preference of HvDMP for certain acyl-CoA chain lengths was assessed in a second in vivo experiment where we supplemented yeast expressing the enzyme with different fatty acid homologs. Preliminary tests showed that co-feeding of $C_{16}$ 3-ketoacid with either $C_{11}$ or $C_{17}$ fatty acids yielded the same β-diketones as controls lacking exogenous acids. However, co-feeding of $C_{16}$ 3-ketoacid with a mixture of $C_{12}$–$C_{15}$ fatty acids led to production of $C_{27}$–$C_{31}$ β-diketones (Fig. 6D). The products had a near-symmetric chain length distribution centered at $C_{29}$, including substantial amounts of two even-numbered β-diketones, $C_{28}$ and $C_{30}$ (Fig. 6E). In contrast to the controls lacking exogenous fatty acids, unsaturated β-diketones could not be detected in this assay. Fatty acid profiling of the total yeast lipids confirmed substantially increased amounts of $C_{12}$–$C_{15}$ acyls in the complemented assays (Fig. 6F). Overall, HvDMP thus condensed $C_{16}$ 3-ketoacid with fatty acyls ranging from $C_{12}$ to $C_{16}$, but preferring $C_{13}$/$C_{14}$ acyls (Supplementary Fig. S8B).

Finally, the interdependence between the HvDMP 3-ketoacid and acyl substrate chain length preferences was explored using in vitro assays with defined, limiting substrate concentrations. In one set of experiments, recombinant HvDMP was incubated with a mixture containing one molar equivalent each of $C_{14}$, $C_{16}$, and $C_{18}$ ketoacids and 1.5 equivalents of a particular fatty acyl-CoA ($C_{12}$, $C_{14}$, or $C_{16}$). In all

three assays, over 60% of the β-diketone products were formed by incorporation of $C_{14}$ 3-ketoacid substrate, accompanied by 20% of $C_{16}$ 3-ketoacid condensation products and circa 10% of $C_{18}$ ketoacid products (Fig. 6G, H). The HvDMP enzyme, thus, had a strong preference for $C_{14}$ 3-ketoacid, irrespective of acyl-CoA co-substrate chain length.

In a final set of experiments, the acyl-CoA preference of HvDMP was tested in competition assays with limited amounts of 3-ketoacid co-substrates, where the enzyme was incubated with a mixture of $C_{12}$, $C_{14}$ and $C_{16}$ fatty acyl-CoAs (1:1:1 molar ratio) and either $C_{14}$, $C_{16}$, or $C_{18}$ 3-ketoacid (1.5 molar equivalents). In all three assays, $C_{16}$ fatty acyl-CoA substrate was slightly preferred over $C_{14}$ fatty acyl-CoA, and less than 7% of the β-diketones originated from $C_{12}$ acyl-CoA (Fig. 6I, J). Overall, the in vitro assays thus showed that HvDMP preferentially condensed $C_{14}$ 3-ketoacid with $C_{14}$ or $C_{16}$ fatty acyl-CoA to form $C_{27}$ or $C_{29}$ 14,16-diketone, respectively.

## Subcellular compartmentation of the β-diketone-forming enzymes

Based on our biochemical results showing that only two enzymes, HvDMH and HvDMP, were required for β-diketone formation, we aimed to localize the biosynthetic pathway across the cellular compartments. For this, we first determined the subcellular localization of HvDMH by confocal microscopy of barley protoplasts transiently expressing green fluorescent protein (GFP) fusion constructs. The resulting GFP fluorescence signal matched the chlorophyll autofluorescence patterns within the cells (Fig. 7A–D), indicating that this enzyme resided in the chloroplast. In contrast, transient expression of GFP-HvDMP fusion proteins in tobacco leaves showed a reticulate pattern and colocalized with ER-specific marker HDEL-RFP (Fig. 7E–G), suggesting that the HvDMP protein resides in the ER.

## Discussion

The major findings of our experiments were that (i) $C_{16}$ 3-ketoacid is the central intermediate en route to barley β-diketones, (ii) DMH generates fatty acid derivatives with functional groups on C-3, and (iii) DMP uses 3-ketoacids and fatty acyl-CoAs for head-to-head condensation into β-diketones. These results can now be integrated to describe the overall biochemistry of the β-diketone biosynthesis pathway.

### Characterization of the first pathway step

Initial information on the β-diketone-forming pathway came from our barley wax analyses. First, we confirmed previous reports that barley spike and flag leaf sheath waxes contained mainly $C_{15}$ 2-alkanol esters[18,27]. Since wax 2-alkanols are thought to be derived from 3-ketoacids one carbon longer[24], this finding suggests that the $C_{16}$ 3-ketoacid is a major precursor available in planta. Secondly, we also confirmed earlier results on the composition of β-diketones with odd carbon numbers[8] and further identified unbranched, even-numbered

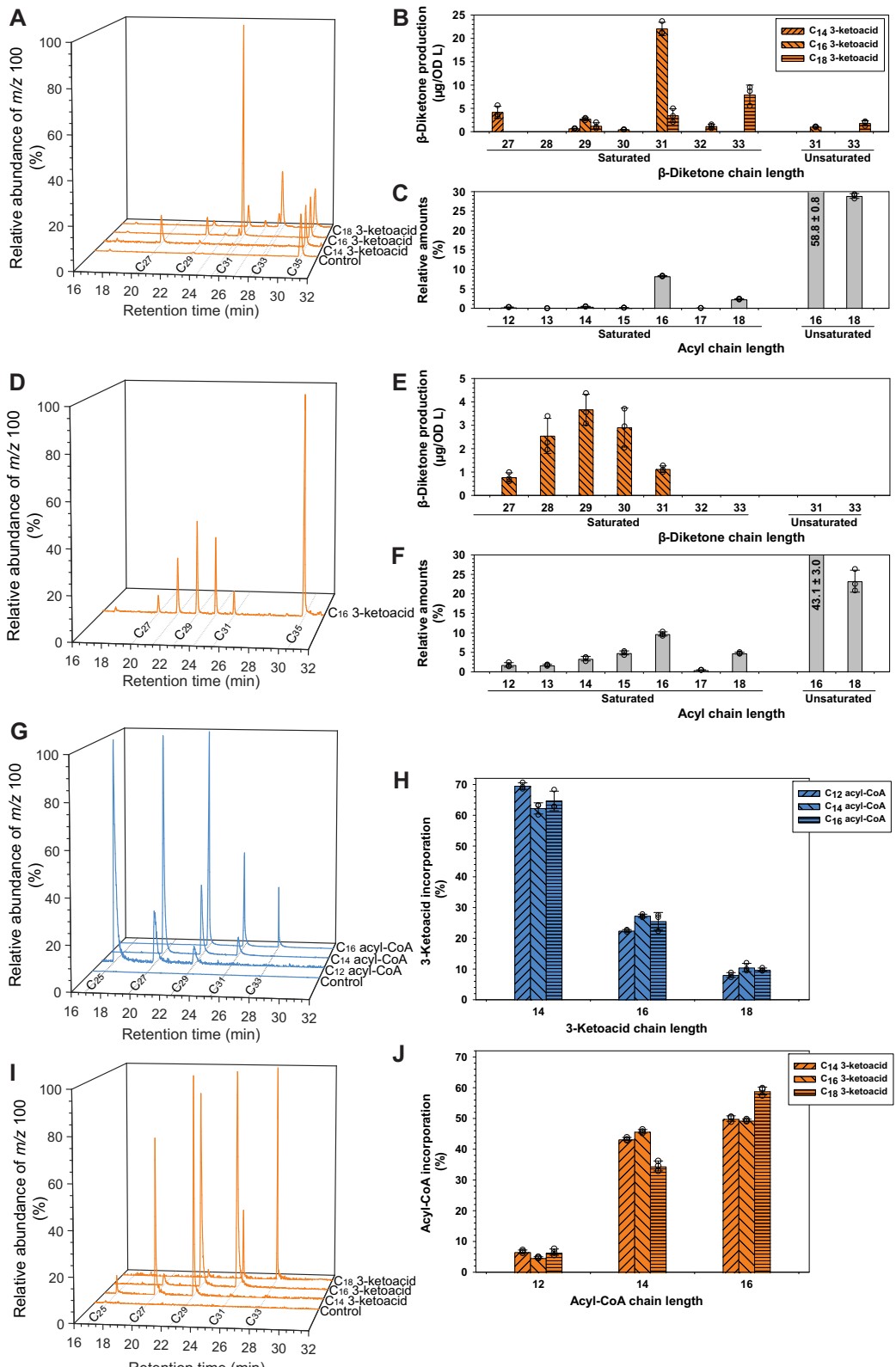

β-diketones. The latter are distinguished by one hydrocarbon tail that can only be formed by elongation from an initial $C_3$ (rather than $C_2$) moiety, and thus must have been present before functional group formation. In particular, our MS analyses showed that these β-diketones have functional groups on C-15/C-17 and C-17/C-19, implying that respective $C_{15}$, $C_{17}$, or $C_{19}$ moieties must be synthesized prior to the diketo group. This left two scenarios, where the odd-numbered

acyls may be either $C_{17/19}$ 3-ketoacyl or $C_{15/17}$ acyl intermediates (see Fig. 1B). Interestingly, the latter scenario implies head-to-head condensation with $C_{16}$ 3-ketoacid and, thus, fits our previous finding that $C_{16}$ 3-ketoacid is the major pathway intermediate. This also matches earlier reports that feeding $C_{15}$ acyl substrate to barley tissue slices led to the formation of $C_{30}$ 14,16-dione[8], one of the even-numbered β-diketones identified here in barley wax.

**Fig. 6 | Assessment of HvDMP substrate preferences.** GC-MS analysis of in vivo and in vitro assays feeding several acyl-CoA and/or ketoacid substrates to HvDMP. Chromatograms of fragments $m/z$ 100 selectively reporting β-diketones are shown on the left and corresponding β-diketone product profile(s) on the right. **A** Selected-ion chromatograms of lipids from yeast expressing *HvDMP* and supplemented with $C_{14}$, $C_{16}$, or $C_{18}$ 3-ketoacid, or only buffer. **B** β-diketone profiles produced in (**A**) (quantified against internal standard $C_{35}$ 14,16-diketone). **C** Relative composition of fatty acyl pools in yeast expressing *HvDMP* (quantified after transmethylation of total lipid extracts of yeast not supplemented with ketoacid). **D** Selected-ion chromatogram of lipids from yeast expressing *HvDMP* and supplemented with $C_{16}$ 3-ketoacid alongside $C_{12}$–$C_{15}$ fatty acids (to adjust the fatty acyl profile). **E** β-diketone profiles produced in (**D**) (quantified against internal standard $C_{35}$ 14,16-diketone). **F** Relative composition of fatty acyl pools in yeast expressing *HvDMP* (quantified after transmethylation of total lipid extracts of yeast

supplemented with $C_{12}$ to $C_{15}$ fatty acids). **G** Selected-ion chromatograms of lipids from HvDMP in vitro assays combining equal molar concentration of $C_{14}$–$C_{18}$ 3-ketoacids alongside $C_{12}$, $C_{14}$ or $C_{16}$ fatty acyl-CoA. Boiled recombinant enzyme incubated with equimolar concentrations of $C_{14}$–$C_{18}$ 3-ketoacids alongside $C_{16}$ fatty acyl-CoA served as control. **H** Proportions of 3-ketoacids incorporated with each fatty acyl-CoA co-substrate into the β-diketones shown in (**G**). **I** Selected-ion chromatograms of lipids from HvDMP in vitro assays combining equal molar concentration of $C_{12}$–$C_{16}$ fatty acyl-CoAs with $C_{14}$, $C_{16}$, or $C_{18}$ 3-ketoacid. Boiled recombinant enzyme incubated with equimolar concentrations $C_{12}$–$C_{16}$ fatty acyl-CoAs and $C_{16}$ 3-ketoacid served as control. **J** Proportions of fatty acyl CoAs incorporated with each 3-ketoacid co-substrate into the β-diketones shown in (**I**). Arabidopsis *LACS1* was expressed in all the yeast assays to enhance exogenous substrate uptake[61]. Data are presented as means ± standard deviations of three biological replicates. Source data are provided as a Source Data file.

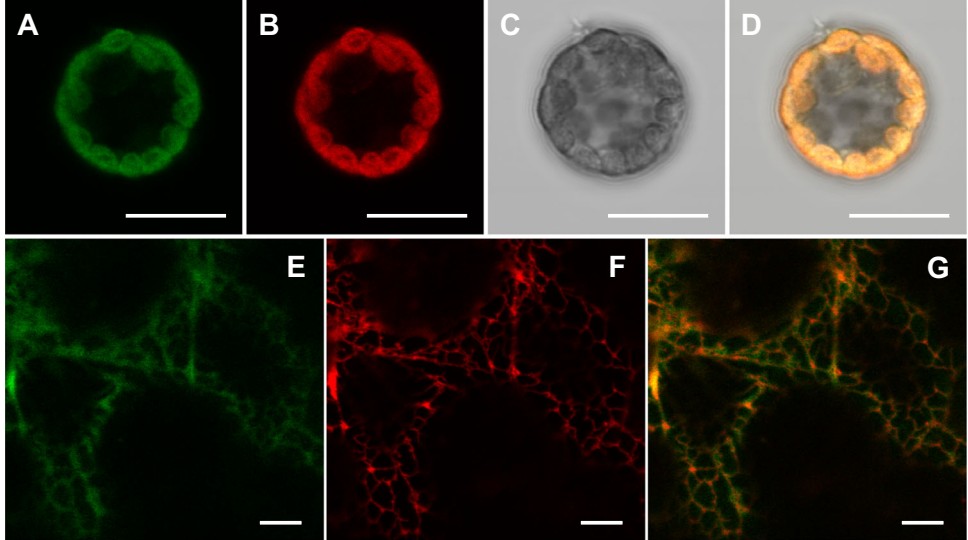

**Fig. 7 | Subcellular localization of HvDMH and HvDMP.** Confocal microscope images of barley protoplasts transiently expressing *Pro35S:HvDMH-GFP*, showing **A** HvDMH-GFP, **B** chloroplast autofluorescence, **C** bright field, and **D** the merged image of (**A**–**C**). Confocal microscope images of *N. benthamiana* leaves coexpressing, **E** *Pro35S:GFP-HvDMP*, and **F** the ER marker *Pro:35S:HDEL-RFP*, and **G** merged image of (**E**, **F**). Bars = 10 μm. Both experiments were independently repeated three times and 10–15 samples/regions were investigated each time with similar results.

Our chemical datasets were complemented by biochemical assays with HvDMH, the enzyme catalyzing the first committed step of the β-diketone pathway[19,20]. Experiments using *E. coli* heterologous expression confirmed the thioesterase activity of HvDMH, and the overall product profiles of HvDMH (in comparison with the acyl pools present) showed that the enzyme had a strong preference for $C_{14}$ and $C_{16}$ acyl substrates. The thioesterase activity of HVDMH was further corroborated using coupled in vitro assays with $C_{16}$ 3-ketoacyl-ACP substrate. The biochemical assays, therefore, underscored the conclusion that $C_{16}$ 3-ketoacid was the major intermediate incorporated into the various β-diketone homologs and isomers.

It is of note that HvDMH expression in *E. coli* yielded high amounts of unsaturated 2-ketones, 2-alkanols, and 3-hydroxyacids, showing that the enzyme does not discriminate between substrates with specific geometry of aliphatic tails including C = C double bonds. Our results suggest that the unsaturated 2-ketones are formed from ω−7 monounsaturated 3-ketoacids which are direct hydrolysis products of corresponding acyl-ACP intermediates of *E. coli* fatty acid synthesis[42]. However, plants typically produce much smaller proportions of unsaturated $C_{16}$ acyls than *E. coli*[43], so that HvDMH will likely encounter lower concentrations of this substrate in plant epidermal cells than in our bacterial assays. The details of our heterologous expression experiment can, therefore, not be directly extrapolated to infer the in planta activity of HvDMH.

In contrast to the HvDMH assays in *E. coli*, HvDMH upon expression in yeast failed to produce 3-ketoacyl intermediates or their derivatives under the tested conditions. It seems plausible that the plant enzyme, HvDMH, may successfully intercept intermediates only from type-II FAS complexes present in plants and *E. coli*[44], but not the type-I FASs present in yeast[45]. However, it is also possible that the hydrolase requires specific ACP isoforms as substrates[46] and therefore discriminates against yeast ACPs.

Altogether, our wax analyses and HvDMH assays firmly established several homologous 3-ketoacids as key intermediates of the β-diketone biosynthesis pathway, with $C_{16}$ 3-ketoacid as the major substrate for ensuing reactions. However, our data thus far did not allow us to distinguish between the elongation and head-to-head condensation hypotheses, and to test these alternatives we investigated the second pathway reaction catalyzed by HvDMP.

**Characterization of the second pathway step**

We initially sought to test the elongation hypothesis with chemical evidence. To this end, we first analyzed the waxes of the barley *emr1* mutant deficient in the condensing enzyme of the FAE complex, KCS6/CER6[35]. However, the mutant had β-diketone amounts and profiles similar to those of the wild type, suggesting that β-diketone biosynthesis must either involve an elongation machinery largely independent of that forming all other wax compounds or else may not rely on ER-based

elongation mechanisms at all. This conclusion confirmed previous reports that β-diketone biosynthesis is not influenced by hydrocarbon elongation inhibitors[25]. Second, we gauged the portions of carbon atoms incorporated in the ER, the compartment known to harbor all VLC elongation enzymes[2], against carbons incorporated in the plastids during initial fatty acyl assembly. Our results revealed a positive correlation of $^{13}$C isotope values with plastidial carbon proportions across all barley wax compound classes with known biosynthesis mechanisms, similar to trends reported before for some but not all wax mixtures of other species[39,47,48]. The $^{13}$C value of barley wax β-diketone very closely matched that predicted assuming that all its carbons are of plastidial origin and clearly differed from that predicted for ER elongation of a $C_{16}$ precursor. The isotope evidence, thus, favored the head-to-head condensation hypothesis for the second step of the β-diketone biosynthesis pathway over the elongation hypothesis.

To test the head-to-head condensation hypothesis in detail, we assayed the activity of the key enzyme of the β-diketone pathway, HvDMP. Feeding of $C_{16}$ 3-ketoacid to yeast expressing *HvDMP* led to the formation of $C_{31}$ 14,16-diketone and, thus, showed overall condensation activity. Corresponding yeast assays supplemented by deuterium-labeled or odd-numbered fatty acids unambiguously showed that the entire chain of the acyl substrate was incorporated into the β-diketones. Similarly, feeding of characteristic 3-ketoacid and fatty acid further confirmed that both precursors were directly condensed into the product. These findings were finally corroborated by in vitro assays where HvDMP incorporated $C_{16}$ 3-ketoacid and $C_{16}$ fatty acyl-CoA directly into $C_{31}$ 14,16-diketone. Together, our experiments thus characterized HvDMP as a condensing enzyme utilizing long-chain acyl-CoA starter and 3-ketoacid extender, in lieu of the aromatic acyl-CoA starters and malonyl-CoA extenders of most other plant PKSs[28,44,45].

Only few non-canonical plant PKSs had previously been shown to catalyze reactions somewhat similar to HvDMP (Supplementary Fig. S9). For example, *Curcuma longa* CURSs condense a coumaroyl-CoA or feruloyl-CoA starter with a coumaroyldiketide or feruloyldiketide acid extender[32,49], while the *Aquilaria sinensis* phenylethylchromone-forming polyketide synthase (PECPS) condenses benzoyl-CoA starters with aromatic extender[50]. In contrast, the alkylquinolone synthase (AQS) from *Tetradium ruticarpum* (formerly *Euodia ruticarpa*) combines a characteristic aromatic starter, *N*-methylanthraniloyl-CoA, with a medium-chain 3-ketoacid en route to alkaloid products[51]. Similar activities involving aromatic starters and aliphatic extenders were also shown, albeit only in vitro, for rice (*Oryza sativa*) curcuminoid synthase (CUS)[52,53], multiple PKSs from *A. sinensis* and a *Huperzia serrata* PKS (HsPKS)[54–56]. Overall, these enzymes catalyze comparable condensation reactions not involving malonate but instead using extenders similar to their (mostly aromatic) starters. HvDMP also condenses two substrates that are similar to each other, however both are long-chain compounds very different from those of the other PKSs.

Interestingly, HvDMP shares 55% similarity with CUS, 42% with CURS, 42% with AQS, 37% with HsPKS3, 44% with PECPS and 44% with *Medicago sativa* CHS[32,50–52,55,57], and sequence comparisons clearly identify a Cys-His-Asn catalytic triad in HvDMP essential for plant PKSs[31]. Accordingly, we propose a reaction mechanism analogous to other condensing enzymes[53], proceeding via initial covalent binding of the starter acyl, entry of the ketoacid extender and decarboxylative Claisen condensation between both substrates (Supplementary Fig. S10). It will be interesting to determine the tertiary structure of HvDMP, for comparison of its active site cavity with those of plant PKSs using substrates with very different molecular geometry and polarity.

## Context of the β-diketone biosynthesis pathway

Finally, the structures of the β-diketones accumulating in barley waxes can be explained based on the biochemistry of the two key enzymes

and a comparison of the substrate preferences of both pathway steps. HvDMP prefers $C_{14}$ 3-ketoacid substrate but also readily accepts the $C_{16}$ 3-ketoacid mainly produced by HvDMH. Similarly, HvDMP has high activity on saturated $C_{14}$ fatty acyl co-substrates, but also on the $C_{16}$ fatty acyls predominating its substrate pool in planta[58,59]. Both pathway enzymes, thus, have sufficient activity on their respective $C_{16}$ substrates to make the preponderance of this chain length in the acyl pool translate into its strong dominance in the final β-diketone homolog and isomer composition. This explains why the one β-diketone generated from $C_{16}$ acyl and $C_{16}$ 3-ketoacyl moieties, $C_{31}$ 14,16-diketone, makes up more than 96% of the product mixture in barley.

The 3-ketoacid generated by HvDMH in plastids must be exported to the ER, the subcellular compartment of the HvDMP, and the 3-ketoacids must be chemically stable enough to survive this transport process (likely occurring by diffusion of the lipophilic compound in respective bilayer membranes). In this context, it is of note that the 3-ketoacids proved fairly stable at room temperature in our in vivo and in vitro assays, but were prone to decarboxylation at elevated temperatures as previously reported[19]. The second substrate required by HvDMP, the long-chain fatty acyl, is also formed in the plastids and exported to the ER, with CoA thioesterification occurring in the process[2,60]. Assembly of β-diketones then occurs by HvDMP, 3-ketoacid and acyl-CoA all coming together in the ER, where also all other wax compounds are synthesized[2]. However, the formation of the other wax constituents there proceeds in small increments gradually increasing the lipophilicity of the compounds, whereas β-diketone synthesis is accomplished in a single step going directly from long-chain precursors to a highly lipophilic VLC hydrocarbon structure.

Taking all our findings together, we propose a model for the biosynthesis of β-diketones and their associated wax compounds, where (i) HvDMH acts as a thioesterase that intercepts 3-ketoacyl-ACPs from type-II FAS complexes to produce 3-ketoacids in plastids of epidermis cells; (ii) the 3-ketoacids are then exported to the ER, likely through bilayer membrane contact sites and/or hemifusions; (iii) fatty acyl-CoAs derived from plastidial fatty acid de novo synthesis are also imported into the ER; (iv) in the ER, the DMP enzyme catalyzes a decarboxylative head-to-head condensation of both substrates to form β-diketones; (v) most of the β-diketones are exported directly to the plant surface, while (vi) some are, likely still in the ER, first hydroxylated to hydroxy-β-diketones by HvDMC and then exported (Fig. 1A). In a side reaction, part of the 3-ketoacid intermediates may be decarboxylated either spontaneously or by a decarboxylase into 2-ketones, which are reduced to 2-alkanols and esterified with fatty acyl-CoAs to form 2-alkanol esters. It cannot be ruled out that the 2-alkanols are formed by spontaneous or enzymatic decarboxylation of 3-hydroxyacids (originating from HvDMH-catalyzed hydrolysis of 3-hydroxyacyl-ACPs).

Since the formation of the β-diketones and their derivatives recruits substrates directly from fatty acid synthesis, it must compete with co-occurring processes of primary lipid metabolism. Therefore, β-diketone biosynthesis must be tightly regulated, especially during the expansion of epidermal cells in the course of organ growth. In this context, it is of particular interest that the β-diketone biosynthesis genes are clustered in the barley and wheat genomes[19,20,23], and their expression is under tight control by inhibitory elements in wheat[23]. It will be interesting to investigate the regulatory mechanisms in the context of primary lipid metabolism. It will also be interesting to compare mechanisms of β-diketone formation between barley and diverse other species known to also accumulate these compounds in their surface waxes.

In conclusion, the evidence provided here characterizes HvDMP as a PKS condensing non-aromatic acyl-CoA starters with unusual ketoacid extenders to directly form the β-diketones accumulating in barley leaf and spike surface waxes. The unique ketoacid substrates are generated by a specific thioesterase, HvDMH, intercepting

intermediates of fatty acid biosynthesis in the plastid. The homolog and isomer distribution of the barley wax β-diketones are fully explained by the combined chain length preferences of both enzymes and the available fatty acyl pool. The biosynthesis pathway to β-diketones is, thus, very distinct from pathways leading to other wax compounds found ubiquitously in the wax mixtures of plants. Only two enzymes define the entire pathway to β-diketones and their derivatives, in stark contrast to the previously assumed multi-step process relying on incremental elongation of the β-diketone carbon backbone. These insights into the biosynthesis of major components of Poaceae cuticles will enable further research towards breeding of cereal crop lines with improved drought tolerance.

## Methods

### Chemicals

All chemicals were obtained from Sigma-Aldrich, unless specified otherwise below.

### Plant materials and growth conditions

*H. vulgare* cv. Morex (NGB23015) seeds were obtained from the Nordic Genetic Resource Center (Alnarp, Sweden), seeds of *H. vulgare* cv. Ingrid and *emr1* were kindly provided by Dr. U. Schaffrath (Technische Hochschule Aachen, Germany). Seeds were germinated on moist filter paper in Petri dishes at room temperature, then transferred to soil (Sunshine Mix 5, Sun Gro) in 10-l pots. Plants were grown in a growth chamber under 16-h light/21 °C, 8-h dark/19 °C cycles. For barley protoplast preparation, barley cv. Morex seeds were germinated on moist filter paper in Petri dishes, and plantlets were transferred to soil in small pots covered with plastic domes and kept in the growth chamber for 10 days under the conditions described above. Tobacco (*Nicotiana benthamiana*) plants grown under the same conditions for 1 month were used for leaf transient expression.

### RNA isolation, reverse transcription, plasmid construction, and transformation

Flag leaf sheaths and spikes of *H. vulgare* cv. Morex were excised and immediately frozen in liquid nitrogen, total RNA was extracted using PureLink RNA mini kit (Invitrogen), and genomic DNA was removed by on-column DNA digestion with PureLink DNase (Invitrogen) following the manufacturer's protocol. In total, 1 μg of resulting total RNA and Oligo(dT)$_{20}$ primer (Invitrogen) was used to synthesize first-strand complementary DNA (cDNA) with SuperScript III Reverse Transcriptase (Invitrogen). The resulting cDNA served as a template for gene cloning.

*E. coli* expression vectors pET28-HvDMH, pET28-HvDMP and pET28-TEVH were kindly provided by Dr. A. Aharoni (Weizmann Inst., Rehovot, Israel). The expression cassette contains an N-terminal 6xHis-tag followed by a TEV cleavage site and the open reading frame (ORF) of the target gene. To construct *E. coli* expression vectors pET28a-EcACP, pET28a-EcFabD and pET28a-MtFabH, EcACP was amplified from *E. coli* (DH 5α) with Phusion polymerase (NEB) using primers EcACP-NdeI-F/EcACP-BamHI-R (Supplementary Table S4). The EcFabD and MtFabH sequences were synthesized by Twist Bioscience and further amplified using primers EcFabD-NdeI-F/EcFabD-BamHI-R and MtFabH-NdeI-F/MtFabH-BamHI-R, respectively. The PCR products were purified and inserted into pET28a vector and in frame with N-terminal 6xHis-tag followed by a thrombin cleavage site. *E. coli* transformed with target vector were screened on lysogeny broth (LB) plates containing 50 μg/ml kanamycin, and verified by colony PCR, restriction digestions, and Sanger sequencing. The confirmed expression vectors were transformed into *E. coli* BL21 (DE3) competent cells (Invitrogen) according to the manufacturer's protocol, and the presence of the gene insert was verified by colony PCR. Three to four independent transformants of each *E. coli* line were selected for further in vivo assay or protein purification.

To construct plant expression vectors, the HvDMH and HvDMP coding regions were amplified from *H. vulgare* cv. Morex flag leaf sheath cDNA with primers HvDMH-F/HvDMH-R and HvDMP-F/HvDMP-R, respectively (Supplementary Table S4). PCR products were ligated to Gateway entry vector pCR8/GW/TOPO (Invitrogen) according to the manufacturer's protocol, and the presence of the target sequences was verified by colony PCR and Sanger sequencing. LR reactions were performed with LR Clonase II enzyme mix (Invitrogen) to transfer target genes from the entry vector to the destination vector for in-frame C-terminal or N-terminal fusion with GFP. Positive transformants were identified by colony PCR and Sanger sequencing. The resulting constructs were transformed into *Agrobacterium tumefaciens* GV3101 competent cells by electroporation, screened on LB plates containing proper antibiotics, and verified by colony PCR. Three verified transformants were cultured for tobacco transient expression.

To construct yeast expression vectors, the coding sequence of long-chain acyl-CoA synthetase 1 (LACS1) that enhances yeast lipid uptake[61] was amplified from Arabidopsis stem cDNA using primers LACS1-BamHI-F and LACS1-SalI-R and inserted into the pESC-Trp yeast expression vector in-frame with C-terminal MYC epitope tag, for expression under control of the galactose-inducible promoter GAL1, resulting in construct pESC-Trp-GAL1::LACS1-MYC. HvDMP was subcloned from pGWB6-35Spro::GFP-HvDMP using primers HvDMP-EcoRI-F and HvDMP-SpeI-R and spliced into pESC-Trp-GAL1::LACS1-MYC vector, C-terminally in-frame with FLAG epitope tag, resulting in construct pESC-Trp-GAL1::LACS1-MYC-GAL10::HvDMP-FLAG. *E. coli* transformed with either vector were screened on LB plates containing 100 μg/ml ampicillin, and verified by colony PCR, restriction digestions and Sanger sequencing. The resulting expression vectors were transformed into yeast strain INVSc1 (*MATα/MATa his3Δ1/ his3Δ1 leu2/leu2 trp1-289/trp1-289 ura3-52/ura3-52*) as described[62], and the resulting transformants were screened on yeast minimal medium plates lacking the appropriate amino acids. The transformed yeast colonies were verified by colony PCR. Three to four verified yeast transformants were selected for further in vivo assays.

### Chemical synthesis of standards and substrates

**C$_{31}$ 14,16-diketone (hentriacontane-14,16-dione).** In all, 1 g of hexadecanol was stirred at room temperature with 2.5 g of Dess-Martin reagent periodinane in 30 ml CH$_2$Cl$_2$. Upon completion of the oxidation (monitored by TLC), the reagent was quenched with aqueous Na$_2$S$_2$O$_3$ solution. The mixture was extracted with CH$_2$Cl$_2$, and the combined organic fractions were washed with aqueous NaH$_2$CO$_3$ solution, dried with Na$_2$SO$_4$ and concentrated in vacuo. The product, hexadecanal, was purified by column chromatography (CHCl$_3$) and verified by GC-MS analysis.

In all, 0.7 g 2-pentadecanone was stirred at 0 °C with 0.1 M lithium diisopropylamide in 11.5 ml tetrahydrofuran (THF) for 10 min, then 2 ml THF containing 0.8 g hexadecanal were added drop-wise, and after 1 h the reaction was quenched with 5 ml 20% H$_2$SO$_4$. The mixture was extracted with CH$_2$Cl$_2$, and the combined organic solutions were washed, dried and concentrated described as above. The product, 16-hydroxyhentriacontan-14-one, was purified by column chromatography (1:1 hexane:CHCl$_3$) and verified by GC-MS analysis.

In all, 0.5 ml 1 M oxalyl chloride in CH$_2$Cl$_2$ was added drop-wise to 150 μl dimethyl sulfoxide (DMSO) in 2 ml CH$_2$Cl$_2$ under stirring at −78 °C. After gas evolution had ceased, 0.2 g 16-hydroxyhentriacontan-14-one in 3 ml CH$_2$Cl$_2$ were added drop-wise, the reaction mixture was allowed to warm to room temperature, and then 0.3 ml Et$_3$N was added. The resulting solid was re-dissolved by adding 5 ml water, and organic products were extracted with CH$_2$Cl$_2$. The combined organic phases were washed first with diluted HCl and then with water, dried with Na$_2$SO$_4$, and concentrated. The resulting product, hentriacontane-14,16-dione, was purified by column chromatography (3:1 hexane: CHCl$_3$) and verified by MS analysis.

Other diketone homologs were synthesized following the same protocol from starting materials with respective chain lengths.

**C$_{21}$ 2-alkanol (2-heneicosanol).** In total, 91 mg of LiAlH$_4$ were added to a solution of 500 mg 2-heneicosanone in 10 ml CH$_2$Cl$_2$. The reaction was quenched after 1 h by slowly adding 2 ml of 1 N HCl, and products were recovered by extraction with CHCl$_3$. Organic fractions were combined, washed with saturated aqueous NaH$_2$CO$_3$, dried with Na$_2$SO$_4$, and concentrated under vacuum. The final product was purified by column chromatography (CHCl$_3$).

**C$_{14}$–C$_{18}$ 3-ketoacids.** C$_{16}$ 3-ketoacid methyl ester was synthesized as described previously[63], with small modifications: 0.45 g myristic acid was dissolved in 3 ml CH$_2$Cl$_2$ containing 0.4 g N,N'-dicyclohexylcarbodiimide. Separately, 1.9 g 4-dimethylaminopyridine was added to 20 ml pyridine containing 1.6 g Meldrum's acid. Both mixtures were stirred at room temperature for 15 min before combining them. After overnight stirring, the solvent was evaporated under vacuum and the solid re-dissolved in 20 ml methanol supplemented with two drops of concentrated H$_2$SO$_4$. The mixture was refluxed overnight, and product was extracted with Et$_2$O. The combined organic fractions were washed with aqueous NaHCO$_3$ solution, distilled water and finally saturated NaCl, the organic phase removed and dried with Na$_2$SO$_4$, and the solvent evaporated in vacuo. The resulting solid was purified by column chromatography (CHCl$_3$) and product purity verified by GC-MS analysis.

In all, 0.5 g C$_{16}$ 3-ketoacid methyl ester was dissolved in glacial acetic acid supplemented with concentrated HCl, and the mixture stored at 10 °C. Prior to each experiment, product was obtained by extraction with Et$_2$O. The organic phase was washed with deionized water, and the solvent evaporated under vacuum. GC-MS analysis showed that the crude product contained non-hydrolyzed C$_{16}$ 3-ketoacid methyl ester and C$_{16}$ 3-ketoacid as main products, accompanied by small amounts of the decarboxylation product, C$_{15}$ 2-ketone. This mixture was separated by flash column chromatography (CHCl$_3$), to yield pure C$_{16}$ 3-ketoacid for use in biochemical assays (while C$_{16}$ 3-ketoacid methyl ester was recovered for further hydrolysis).

The same protocols were used to synthesize C$_{14}$, C$_{17}$, and C$_{18}$ 3-ketoacids.

## Enzyme assays

***E. coli* in vivo assay.** Three independent *E. coli* BL21 (DE3) colonies each carrying pET28-TEVH or pET28-HvDMH were inoculated into 1 ml LB liquid medium with kanamycin and precultured at 37 °C overnight. Then, each broth was diluted 1:100 in 100 ml LB liquid medium supplemented with kanamycin, cultivated at 37 °C to OD$_{600}$ ~0.6, supplemented with 0.5 mM isopropyl β-D-1-thiogalactopyranoside (IPTG) to induce the expression of recombinant protein, and incubated at 22 °C overnight. Then, the OD$_{600}$ were measured for normalization against cell numbers and *E. coli* cells collected by centrifugation for lipid analysis.

**Yeast in vivo assay.** Three independent yeast colonies each of lines expressing pESC-Trp-GAL1::LACS1-MYC or pESC-Trp-GAL1::LACS1-MYC-GAL10::HvDMP-FLAG were inoculated into 2 ml appropriate liquid minimal medium supplemented with 2% glucose and incubated at 28 °C overnight. The precultures were then expanded to 30 ml and cultivated for 18 h. The resulting yeast cells were collected by centrifugation and transferred into liquid minimal medium supplemented with 2% galactose to induce the expression of target genes and cultivated at 28 °C for 16 h. Then, yeast cells were transferred to 30 ml liquid minimal medium based on phosphate buffer (20 mM Na$_2$HPO$_4$/NaH$_2$PO$_4$, 300 mM NaCl, pH 7.4) and containing 2% galactose. Enzyme substrates (Supplementary Table S5) dissolved in ethanol were added (0.22 mM final concentrations of fatty acids and 3-ketoacids), and the yeast cells were cultivated for another 24 h at 22 °C. Finally, the OD$_{600}$ of the resulting yeast were measured for normalization against cell numbers, and cells were harvested by centrifugation for further product analysis.

**Protein purification and in vitro assay.** *E. coli* BL21 (DE3) colonies carrying pET28a-EcACP, pET28a-EcFabD, pET28a-MtFabH, pET28-HvDMH or pET28-HvDMP were inoculated into 5 ml LB liquid medium with kanamycin and precultured at 37 °C overnight. *E. coli* precultures were diluted 1:100 in LB liquid medium supplemented with kanamycin and cultivated at 37 °C to OD$_{600}$ ~0.6, and 0.5 mM IPTG was added to induce expression of recombinant protein. After incubating at 16 °C overnight (*E. coli* expressing MtFabH were incubated at 37 °C for 3 h), *E. coli* cells were collected by centrifugation (10,000× *g*, 4 °C). All following steps of protein purification were performed using phosphate-based buffers (20 mM Na$_2$HPO$_4$/NaH$_2$PO$_4$, 300 mM NaCl with imidazole, pH 7.4) supplemented with 1% Triton X-100, and equilibration buffer was supplemented with 0.1 mM phenylmethylsulfonyl fluoride (PMSF) as protease inhibitor. *E. coli* pellets were resuspended in equilibration buffer and lysed by sonication at 4 °C. The recombinant EcACP, EcFabD, MtFabH, HvDMH, and HvDMP proteins were obtained by Ni-NTA affinity purification with HisPur Ni-NTA resin (Invitrogen) following the manufacturer's protocol. Respective recombinant proteins were eluted in phosphate buffer and dialyzed three times for 7 h in dialysis buffer (0.1 M Na$_2$HPO$_4$/NaH$_2$PO$_4$, 2.5 mM dithiothreitol (DTT), 10 mM MgCl$_2$, 0.025% Triton X-100, pH 7.4) at 4 °C. The purity of recombinant proteins was verified by SDS-PAGE, and the protein concentration was determined by protein assay using the Bradford method with Protein Assay Dye Reagent Concentrate (Bio-Rad). The obtained proteins were used immediately in enzyme assays.

HvDMH in vitro assays were performed as described previously[40,41]. In brief, recombinant EcACP (1.0 mg), EcFabD (100 μg) and MtFabH (100 μg) were coincubated in 2 ml 1,3-bis(tris(hydroxymethyl) methylamino propane (20 mM, pH 7.0) containing malonyl-CoA (0.2 mM) and C$_{14}$ acyl-CoA (0.2 mM) at 37 °C for 5 h. 500 μg of HvDMH were added, and the reaction was incubated for an additional 2 h. The reaction was quenched by one drop of 2 mM HCl with tetracosane (3 μg) supplemented as an internal standard.

For HvDMP in vitro assays, 50 μg of purified recombinant protein was incubated in 3 ml 50 mM phosphate buffer (pH 7.4) containing 2.5 mM DTT, 10 mM MgCl$_2$ and 0.025% Triton X-100 with the substrate combinations (Supplementary Table S6) at room temperature for 14 h with gentle shaking. The reaction was quenched by adding 200 μl 10% H$_2$SO$_4$, and the reaction products were extracted immediately for lipid analysis.

## Lipid analysis

**Wax analysis.** Flag leaf sheaths and spikes were excised from *H. vulgare* cv. Morex plants grown 85 d after germination. Surface waxes were extracted by submerging tissues twice for 30 s in CHCl$_3$, extracts were combined, and the solvent was evaporated under gentle N$_2$ flow. Samples were transferred into GC vials and derivatized with 10 μl bis-*N,O*-trimethylsilyltrifluoroacetamide (BSTFA) in 10 μl pyridine at 72 °C for 45 min. The reagents were evaporated under gentle N$_2$ flow, and the product was dissolved in 200 μl CHCl$_3$. The derivatized wax mixtures were then analyzed on a Gas Chromatography system (6890 N, Agilent) equipped with HP-1 capillary column (30 m × 0.32 mm i.d., 1-μm film thickness, Agilent) coupled with Mass Selective Detector (5793 N, Agilent) using cool-on-column injection and the oven program: 50 °C held for 2 min, raised by 40 °C min$^{-1}$ to 200 °C and held for 2 min, raised by 3 °C min$^{-1}$ to 320 °C and held for 30 min.

***E. coli* lipid analysis.** *E. coli* pellets were washed with distilled water, collected by centrifugation and allowed to air-dry. Known quantities of

$C_{21}$ 2-ketone, $C_{21}$ 2-alkanol and $C_{20}$ 2-hydroxyacid methyl ester were added to each sample for quantification of respective compound classes, *E. coli* cells were resuspended in 1 ml methanol containing 0.5 M sulfuric acid and 2% (v/v) 2,2-dimethoxypropane, and incubated at 80 °C for 3 h. Then 2.5% NaCl solution (w/v) was added, and the mixture was extracted three times with hexane, and the combined organic phases were concentrated under $N_2$ flow and derivatized as described above. The final products were dissolved in $CHCl_3$ and analyzed by GC-MS as above, but with the following oven program: 50 °C held for 2 min, raised by 40 °C min$^{-1}$ to 100 °C and held for 2 min, raised by 3 °C min$^{-1}$ to 320 °C and held for 5 min. The correction factor for quantifying 3-hydroxyacids was experimentally determined using authentic $C_{16}$ 3-hydroxyacid and $C_{20}$ 2-hydroxyacid standards.

**Yeast lipid analysis.** Overall, 30 ml-yeast cultures were split into 25 ml for β-diketone analysis and 5 ml for fatty acid profiling, and cells were collected by centrifugation, washed with distilled water, and allowed to air-dry. For product analysis, $C_{35}$ 14,16- diketone was supplemented to yeast pellets as internal standard. Yeast cells were resuspended in 3 ml distilled water and lysed with 0.5 mm glass beads (BioSpec) by vertexing. 2 ml saturated NaCl solution was added, and total lipids were extracted three times with $CHCl_3$, the combined organic phases were dried with $Na_2SO_4$ and filtered, and the solvent was eva-porated under vacuum. The resulting lipids were transferred into GC vials, derivatized, and analyzed by GC-MS as described for wax analy-sis. All diketones were quantified using $m/z$ 100 selected-ion traces. For fatty acid profiling, yeast cells were resuspended in 1 ml methanol containing 0.5 M sulfuric acid and 2% (v/v) 2,2-dimethoxypropane. The transmethylation reaction was performed as described above, and products were extracted with hexane, derivatized as previously described, and analyzed by GC-MS as described for *E. coli* lipid analysis.

**In vitro product analysis.** Samples from HvDMH in vitro assay were extracted three times with hexane, and the combined organic phases were concentrated under $N_2$, derivatized and analyzed by GC-MS as described for wax analysis. Samples of the HvDMP in vitro reaction mixture were prepared in the same way, but extracting twice with $CHCl_3$ and drying under vacuum.

**Carbon isotope analysis**
Wax was extracted from *H. vulgare* cv. Morex flag leaf sheaths as described above and fractionated by TLC ($CHCl_3$: hexane (1:1); 0.25 mm layer thickness; Analtech). TLC bands were stained with pri-muline (1% in acetone) and visualized under UV light, target fractions were scratched from the plate and extracted with $CHCl_3$, and fraction purities were verified by GC-MS analysis. For the two fractions con-taining esters/alkanes and β-diketones, compound-specific carbon isotope compositions were determined using GC-IRMS as described before[64]. In brief, a trace GC system equipped with Rxi-5ms column (30 m × 0.25 mm, film thickness 0.25 μm, Restek) and a programmable temperature vaporizing injector operated in solvent split mode was connected via a GC Isolink to a combustion furnace at 1000 °C and via a Conflo IV interface further to a DeltaVPlus isotope ratio mass spec-trometer (Thermo Scientific). The $CO_2$ reference gas was analyzed daily with a standard deviation of <0.06‰ for instrument stability and linearity. Six reference peaks of $CO_2$ bracketed analyte peaks during the course of a GC-IRMS run; two were used for standardization, the rest were used to monitor stability. Samples were interspersed with an external standard (A6mix) obtained from Dr. A. Schimmelmann, Indi-ana University, Bloomington, containing 15 alkanes ($C_{16}$ to $C_{30}$) with $δ^{13}C$ values ranging from −33.34 to −26.15‰ in order to establish a multi-point correlation, with which data were normalized to the Vienna Pee Dee Belemnite (VPDB) carbon isotopic scale.

**Barley protoplast and tobacco leaf transient expression and microscopy**
Two-week-old *H. vulgare* cv. Morex seedlings were used for protoplast preparation and transformation following the method described previously[65]. pGWB5-35Spro::HvDMH-GFP was used for protoplast transient expression. Transformed protoplasts were incubated for 16–18 h and imaged by an Olympus FV1000 multiphoton confocal laser scanning microscope equipped with 405 nm, 473 nm, and 559 nm lasers. Transient expression of pGWB6-35Spro::GFP-HvDMP and ER-specific marker p35S:HDEL-RFP (obtained from Drs. M. Schuetz and L. Samuels, UBC, Vancouver) were in *N. benthamiana* was performed as described previously[66] and imaged with the same microscope. GFP and RFP were at 488 nm and 561 nm, respectively, and fluorescence of GFP, RFP, and chlorophyll was detected at 525 nm, 595–625 nm, and 660–750 nm, respectively. ImageJ 1.51r software was used for image processing.

**Statistics and reproducibility**
Microsoft Excel (version 2108) and GraphPad Prism 10 software were used for statistical analysis. Two-tailed Student's *t* tests were used to assess significant differences between samples in different assays or wax analyses. Details are provided in the captions of the corresponding figures. All experiments and assays were repeated independently two to three times with similar results.

**Reporting summary**
Further information on research design is available in the Nature Portfolio Reporting Summary linked to this article.

## Data availability
The sequences of the genes used in this article can be found in NCBI GenBank or the Arabidopsis information resource (TAIR) under the following accession numbers: HvDMH (KU721941, MLOC_13397, Cer-q); HvDMP (KU721941, MLOC_59804, Cer-c); EcACP (U00096); EcFabD (U00096); MtFabH (AL123456); LACS1(AT2G47240). The authors declare that all the data supporting the findings of this study are available within this paper and its Supplementary Information file. Source data are provided with this paper.

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

## Acknowledgements

We thank Drs. Mathias Schuetz and Lacey Samuels (Univ. of British Columbia, Canada) for kindly providing the ER-specific marker 35 S:HDEL-RFP, Dr. Asaph Aharoni (Weizman Inst. of Science, Rehovot, Israel) for the vectors pET28-HvDMH, pET28-HvDMP and pET28-TEVH, Dr. Ulrich Schaffrath (Rheinisch-Westfälische Technische Hochschule Aachen, Germany) for seeds of *H. vulgare* cv. Ingrid and *emr1* mutant, the Nordic Genetic Resource Center (Alnarp, Sweden) for seeds of *H. vulgare* cv. Morex, and the Univ. of British Columbia BioImaging Facility for providing microscopes and assistance with confocal imaging. This work has been supported by the Discovery Grants Program of the Natural Sciences and Engineering Research Council (Canada), the Canada Foundation for Innovation, China Scholarship Council Fellowships (to Y.S. and Z.Z.), and the Women in Science and Engineering Program at USC to (S.J.F.).

## Author contributions

Y.S., A.R.O., and R.J. designed the experiments. Y.S. and Z.Z. constructed vectors. A.R.O. synthesized standards and substrates. Y.S., A.R.O., Z.Z., and R.J. performed lipid analyses. S.J.F. determined isotopic compositions. Y.S., A.R.O., Z.Z., and R.J. made the figures. Y.S. and R.J. drafted the text, and all authors edited the text.

## Competing interests
The authors declare no competing interests.
