## [Peer Review File · Nature Communications]

Biosynthesis of barley wax β -diketones: a type-III polyketide synthase condensing two fatty acyl unitsREVIEWER COMMENTS

Reviewer #1 (Remarks to the Author):

Sun et al. reported the discovery of the biosynthetic pathway of barley wax beta-diketones involved in a hydrolase HvDMH and a type III polyketide synthase (PKS), HvDMP. Through in vivo enzyme assays using yeast and in vitro assays, the authors showed that this type III PKS harbors novel beta-diketones-producing PKS activities and HvDPH has been also suggested to be a hydrolase. It is thus strongly speculated that both enzymes are indeed involved in the biosynthesis. The HvDPH activity is also novel as the type III PKS. Overall, the topic of the study is interesting, and the quality of the experimental data is generally good. However, some points should be more reliable to support authors conclusion and additional studies are needed to publish it in Nature Communications.

1. A major weakness of the paper is that the in vivo biochemical function of HvDMP in the plants is not clearly demonstrated. It is known that in vitro activities of plant type III PKSs sometimes differ from their major activities in the heterologous expression system. To support the main claim "HvDMP is involved in the biosynthesis of barley beta-diketides", the authors need to demonstrate HvDMP indeed harbors the claimed activities in the plant. For example, this can be demonstrated by VIGS experiment in barley or its tissue culture, which may include callus, to knockdown expression of HvDMP. The transient expression of HvDMP in *Nicotiana* may be also available for this. This can be also said to HvDMH as well.
2. HvDMH has been concluded to hydrolyze 3-ketoacyl-ACPs to produce 3-ketoacid type. However, it is unclear from the present study only. To support this, the authors need to demonstrate HvDMH indeed harbors the claimed activities in vitro or equivalent methods.
3. Substrate preference of HvDMP: To understand the enzyme mechanism, a basic steady-state kinetic analysis is required. If the authors want to compare the relative activity (%) of each product (or for each substrate), at least, values of the catalytic efficiency (k_{cat}/K_m) or maximal velocity (V_{max}) of each enzyme should be provided. Otherwise, to confirm the contribution of ligand-binding, the authors can provide experimental protein-ligand binding results using Isothermal titration calorimetry (ITC) or Surface Plasmon Resonance (SPR).
4. In the discussion, authors mentioned that "Only two plant PKSs, CURS and CUS, had previously been known to catalyze reaction somewhat similar to HvDMP". However, this description is mistake. As the similar type of type III PKSs, HsPKS3 from *Huperzia serrata*, alkylquinolone synthase (AQS) from *Evodia rutaecarpa*, and diarylpentanoid-producing polyketide synthase (PECPS) from *Aquilaria sinensis* have been also reported. The description and sequential discussion should be corrected.
5. In the proposed HvDMP reaction mechanism, Ser200 and Glu196 have been mentioned as the residues involved in the proton donor for the catalytic Cys. However, to the best of my knowledge, such mechanism has been never proposed to the type III PKS: The His residue in the catalytic triad has been suggested as such residue. To propose this mechanism, more explanation why authors could propose this should be provided.

Reviewer #2 (Remarks to the Author):

The authors identified the β -diketone biosynthesis pathway in barley by biochemical characterizations of two enzymes DMH and DMP from previous identified gene cluster. DMH is a thioesterase producing long-chain (mainly C16), and DMP is a rare polyketide synthase (PKS) condensing the 3-ketoacids with long-chain (mainly C16) acyl-CoAs into β -diketones, which are functionally similar to known curcuminoid synthase. The manuscript is very easy to understand and there is nothing new for natural products biosynthetic field. This significance of this paper is that the waxy natural product is related with the surface coatings of cereal plants barley and the authors demonstrated the details of the PKS mediated key step biosynthesis. Anyway, additional modification may be needed to improve the manuscript as listed below before acceptance.

Major comments

1. Page 4: the description "While this provided first evidence for the enzymatic reaction leading to the key intermediate, it did not provide information on the later steps in the pathway." is not proper. Ref 19 identified that DMP in the gene cluster participated in the β -diketone biosynthesis by gene silencing experiments, and provided information about that DMP showed significant sequence similarity to curcuminoid synthase, which catalyzes the head-to-head condensation of two acyl chains. Please make some suitable changes.
2. Page 10: The C-terminal HDEL enables proteins to be localized in the endoplasmic reticulum (ER). Why use the marker HDEL-RFP, not the RFP-HDEL? In addition, please analyze the localization signal peptide of DMP. Pro35S: GFP-HvDMP was lost.

Minor comments

3. Figure 2A, please provide the complete histogram information and the standard deviation of C31 diketone.
4. Page 17: " μ " in "1 μ g of resulting total RNA" need to be Italic, "N" in "an N-terminal 6xHis-tag followed" need to be Italic.
5. The formats of ref 3, 20, 33, 34, 35, 36, 37, 40 and 50 need to be checked.

Reviewer #3 (Remarks to the Author):

Review of the article Biosynthesis of barley wax β -diketones: a type-III polyketide synthase condensing two fatty acyl units. Sun et al.

This is a well written, well organized, and highly informative paper. The investigation of barley β -diketones was among the first genetic models for the study of plant wax biochemical genetics, and the presence of barley's structurally unique β -diketone class of compounds has been a focus of much interest since the 1970s beginning especially with the work of Penelope von Wettstein-Knowles. Previous publications indicate that the β -diketones themselves are important in plant environmental stress adaptation (especially for plant tolerance to drought, and supra-optimal solar radiation and heat), and thereby constitute an ecologically important class of waxes, and define overall a class of wax compounds with high potential for modification toward improving crop stress tolerance.

The authors apply a straightforward scientific approach that elucidates a novel metabolic pathway being employed by barley (and likely other plants that synthesize β -diketones) for

the synthesis of the β -diketones and its derivatives the hydroxy β -diketones as well as the related compounds 2-alkanol esters. A key finding is that the 3-ketoacids are a central intermediary in these pathways, and that the barley derived enzymes Diketone Metabolism Hydrolase (DMH) and Diketone Metabolism Polyketide Synthase (DMP) are together sufficient to generate the β -diketones from plastidial acyl precursors. The authors have a strong publication record involving the chemical analysis of plant waxes, as well as related metabolic engineering studies, and the methods used in this paper are sound.

The authors provide strong evidence that the biosynthesis of β -diketones and hydroxy β -diketones are produced in a way that it does not utilize the previously described sequential 2-carbon-based elongation steps catalyzed by the Fatty Acid Elongase (FAE) pathway, and that their synthesis does not require the endoplasmic reticulum (ER)-localized β -ketoacyl-CoA synthase (KCS6). The authors demonstrated the *emr1* mutant deficient in KCS6 was able to produce β -diketones with normal chain length, even though other wax constituents on *emr1* were much shorter, and so β -diketone synthesis in barley does not require that same elongation pathway. The authors present evidence that there is not another β -diketone parallel elongation pathway, citing a previous article showing that fatty acyl chain elongation inhibitors do not inhibit elongation of β -diketones and hydroxy β -diketones, indicating that β -diketone synthesis does not involve the previously describes ER based FAE. Further, carbon isotope studies by the authors of this paper using GC coupled with isotope-ratio mass spectrometry revealed that the β -diketones on barley are composed of all plastid generated carbons (enriched in ^{13}C), and not the ER associated carbons wherein only about 50% of the carbons would be plastidial if the chains were further elongated using the FAE pathway mechanism (wherein ^{13}C composition would be low). The authors tested these ^{13}C ratios against other wax classes in barley, providing strong evidence to support a novel synthesis that does not involve ER elongation by FAE.

Instead, the authors provide strong evidence for an alternative and novel very long chain β -diketone synthesis pathway that involves Diketone Metabolism Hydrolase (DMH) that has a thioesterase function and generates fatty acyl derivatives with functional groups on the 3rd carbon, and a Diketone Metabolism Polyketide Synthase (DMP) that performs a decarboxylative condensation of 3-ketoacid acyl chains and fatty acyl-CoA to make the β -diketones. *E. coli* and yeast expression studies were used to demonstrate both HvDMH and HvDMP preferred substrate classes and chain length specificity. It's notable that the yeast system expressing HvDMH could not make the 3-ketoacids substrate for use by HvDMP in synthesis of β -diketones, and so that is why exogenous 3-ketoacids needed to be added to test HvDMP activity. As such, this study does not directly show that HvDMH catalyzes the thioesterase reaction leading to 3-ketoacyls as would be necessary in the plant system for β -diketone production. The authors speculate that HvDMH might not interact properly (as it would in planta) with the yeast FAS complexes, or that it requires specific ACP isoforms and discriminates against yeast ACPs, which are reasonable speculations by these authors. However, HvDMH expressed in *E. coli* did hydrolyze substrates to make acyl products having 3-carbon functional groups, which provided evidence that HvDMH likely generates the 3-ketoacyls in planta. It is known that *E. coli* has very low relative amounts of saturated acyl substrates overall (relative to unsaturated), which may explain why detectable amounts of the specific 3-ketoacyls were not generated by HvDMH in the *E. coli* system.

Studies in the yeast system using deuterated C16 acids showed that HvDMP could condense (with decarboxylation) a deuterated C16 fatty acid with 3-ketoacids to make β -diketones, and so this then shows that HvDMP has the capacity to combine two long chain substrates together. HvDMP's ability to condense fatty acyl-CoAs with ketoacid substrates

was further confirmed in yeast experiments using uncommon odd chain length substrates. Further in vitro/yeast studies tested chain length specificity of HvDMP to find that the HvDMP enzyme prefers the C14 3-ketoacid, and either the C14 or C16 fatty acyl-CoA. But notably HvDMP does in fact readily accept the C16 3-ketoacid. The fact the C16 3-ketoacids and C16 fatty acyl-CoAs are by far the dominant substrate pool available in planta explains why the C31 14,16-diketones are the dominant homologues found on the surface of wild-type barley sheaths, being 96% of the waxes. The functions described here for HvDMP are thus consistent with the β -diketone wax profiles observed on barley.

Finally, the authors localized both HvDMH and HvDMP using confocal microscopy, barley protoplast, and GFP constructs, showing the HvDMH enzyme was localized in the chloroplasts and HvDMP in ER membranes. They proposed a quite reasonable model for substrate transfer (and related reaction mechanisms) in the biosynthetic pathway.

They also describe for the first time unbranched even chain length β -diketones, likely derived from an initial C3 rather than the more common C2 moiety. They then noted that the locations of the functional groups on these even chain β -diketones are consistent with a main finding of this report that the C16 3-ketoacids are a major intermediate in this β -diketone pathway.

Although not necessary for publication in the opinion of this reviewer (as will be described below), additional evidence that HvDMH directly catalyzed synthesis of the 3-ketoacyls in yeast or *E. coli* (or another system) would have helped strengthen the narrative that two enzymes are sufficient for β -diketone synthesis. Notwithstanding, that the *E. coli* studies showed that HvDMH produced closely related 3-carbon modified acyl derivatives does provide good evidence that the HvDMH does in fact generate 3-ketoacyls in planta. Regardless, the presentation of only indirect evidence of HvDMH function in 3-ketoacyl synthesis does not detract from the overall main discoveries of this paper, which is the determination that the HvDMP enzyme generates the β -diketones, and that this is done using a novel reaction that does not involve the well-known sequential C2 FAE elongation steps, but instead uses HvDMP to perform a decarboxylative condensation reaction of two long chain plastidial derived substrates. There are no previous reports of such a condensation reaction of long chain acyl compounds in plants, and this is the first report of the synthesis of very long chain aliphatic compounds that does not involve C2 elongation steps.

The findings here represent novel and remarkable discoveries on their own and are clearly supported by the data and its interpretation.

Some minor edits requested:

Under carbon isotope analysis in methods, lines 2 and 3, recommend simply rewording as "were stained with".

Remove the word "at" in line 15 for Fig 1 caption.

In Fig 2A, the green colors in the legend are a bit difficult to align to the bars. Making the colors more different will improve the ease of interpretation. The colors in Fig 2 C are also difficult to visualize. Point symbols should be enlarged

Reviewer #1 (Remarks to the Author):

Sun et al. reported the discovery of the biosynthetic pathway of barley wax beta-diketones involved in a hydrolase HvDMH and a type III polyketide synthase (PKS), HvDMP. Through in vivo enzyme assays using yeast and in vitro assays, the authors showed that this type III PKS harbors novel beta-diketones-producing PKS activities and HvDPH has been also suggested to be a hydrolase. It is thus strongly speculated that both enzymes are indeed involved in the biosynthesis. The HvDPH activity is also novel as the type III PKS. Overall, the topic of the study is interesting, and the quality of the experimental data is generally good. However, some points should be more reliable to support authors conclusion and additional studies are needed to publish it in Nature Communications.

1. A major weakness of the paper is that the in vivo biochemical function of HvDMP in the plants is not clearly demonstrated. It is known that in vitro activities of plant type III PKSs sometimes differ from their major activities in the heterologous expression system. To support the main claim “HvDMP is involved in the biosynthesis of barley beta-diketides”, the authors need to demonstrate HvDMP indeed harbors the claimed activities in the plant. For example, this can be demonstrated by VIGS experiment in barley or its tissue culture, which may include callus, to knockdown expression of HvDMP. The transient expression of HvDMP in *Nicotiana* may be also available for this. This can be also said to HvDMH as well.

We agree that the *in planta* activity of the enzymes must be addressed and apologize for neglecting this in the original version of the manuscript. Indeed, detailed and repeated analyses of respective barley mutants had already firmly established the central roles of both enzymes, DMH and DMP, in the formation of beta-diketones. Corresponding VIGS experiments independently confirmed the in central role of both enzymes also in beta-diketone biosynthesis in wheat. Given the high genetic similarity of both plant species, the *in planta* function of both enzymes on this pathway had thus been firmly established prior to our work, and there was consequently no need for us to repeat respective mutant analyses. However, it was important to reference the previous work on this, and we have added this information to the Introduction section of our manuscript.

2. HvDMH has been concluded to hydrolyze 3-ketoacyl-ACPs to produce 3-ketoacid type. However, it is unclear from the present study only. To support this, the authors need to demonstrate HvDMH indeed harbors the claimed activities in vitro or equivalent methods.

We thank the reviewer for this suggestion and have since performed the necessary experiments to test the HvDMH activity *in vitro*. A new figure (fig 4) presents the additional results, and a newly added paragraph in the Results section on HvDMH explains the findings.

3. Substrate preference of MvDMP: To understand the enzyme mechanism, a basic steady-state kinetic analysis is required. If the authors want to compare the relative activity (%) of each product (or for each substrate), at least, values of the catalytic efficiency (k_{cat}/K_m) or maximal velocity (V_{max}) of each enzyme should be provided. Otherwise, to confirm the contribution of ligand-binding, the authors can provide experimental protein-ligand binding results using Isothermal titration calorimetry (ITC) or Surface Plasmon Resonance (SPR).

We agree that a kinetic characterization of the enzymes would help to quantify specificities. However, due to the highly lipophilic nature of the substrates, their concentrations in respective assays cannot be

controlled and varied enough for proper measurements of kinetic parameters. Therefore, kinetics of no enzymes handling long-chain aliphatics have been reported to date, and substrate specificities were assessed qualitatively at best. For example, most previous work on long-chain or very-long-chain acyl condensing enzymes tended to report only product specificities but not substrate specificities. In comparison with this precedence, our *in vitro* characterizations and competition assays stretch the limits far beyond the previous methods for assessing relative activities on various substrates. However, we understand that biochemists outside the acyl lipid field may have different expectations regarding enzyme characterization. Therefore, we have toned down our descriptions and interpretations of the various experiments into this, for example by replacing “specificity” with “preference”.

4. In the discussion, authors mentioned that “Only two plant PKSs, CURS and CUS, had previously been known to catalyze reaction somewhat similar to HvDMP”. However, this description is mistake. As the similar type of type III PKSs, HsPKS3 from *Huperzia serrata*, alkylquinolone synthase (AQS) from *Evodia rutaecarpa*, and diarylpentanoid-producing polyketide synthase (PECPS) from *Aquilaria sinensis* have been also reported. The description and sequential discussion should be corrected.

Thanks for pointing this out. We have added a paragraph discussing these enzymes, and illustrated the reactions they catalyze either in planta or in vitro in a new supplementary figure (fig. S9).

5. In the proposed HvDMP reaction mechanism, Ser200 and Glu196 have been mentioned as the residues involved in the proton donor for the catalytic Cys. However, to the best of my knowledge, such mechanism has been never proposed to the type III PKS: The His residue in the catalytic triad has been suggested as such residue. To propose this mechanism, more explanation why authors could propose this should be provided.

Our hypothetical reaction mechanism closely follows those proposed by Morita et al 2010 (PNAS 107, 19778-83) and Morita et al 2019 (JBC 294, 15121-36) for the CUS enzyme, since it catalyzes a head-to-head condensation fairly similar to ours. We understand that, in their model, the His in the catalytic triad is assumed to protonate the CoA leaving group of the starter acyl and to deprotonate the catalytic cysteine after the reaction cycle. However, the re-protonation of the catalytic (via water) at the end of the catalytic cycle had in many cases not been shown for related enzyme mechanisms. It is plausible to invoke a combination of nearby Ser/Glu for this proton flow, but we agree that this part of our original mechanism was highly speculative. To avoid confusion, we deleted respective details of the mechanism, and the new version now matches the mechanisms proposed in various previous papers to the best of our knowledge.

Reviewer #2 (Remarks to the Author):

The authors identified the β -diketone biosynthesis pathway in barley by biochemical characterizations of two enzymes DMH and DMP from previous identified gene cluster. DMH is a thioesterase producing long-chain (mainly C16), and DMP is a rare polyketide synthase (PKS) condensing the 3-ketoacids with long-chain (mainly C16) acyl-CoAs into β -diketones, which are functionally similar to known curcuminoid synthase. The manuscript is very easy to understand and there is nothing new for natural products

biosynthetic field. This significance of this paper is that the waxy natural product is related with the surface coatings of cereal plants barley and the authors demonstrated the details of the PKS mediated key step biosynthesis. Anyway, additional modification may be needed to improve the manuscript as listed below before acceptance.

Major comments

1. Page 4: the description “While this provided first evidence for the enzymatic reaction leading to the key intermediate, it did not provide information on the later steps in the pathway.” is not proper. Ref 19 identified that DMP in the gene cluster participated in the β -diketone biosynthesis by gene silencing experiments, and provided information about that DMP showed significant sequence similarity to curcuminoid synthase, which catalyzes the head-to-head condensation of two acyl chains. Please make some suitable changes.

We have added text to the Introduction passage focusing on the sequence of steps on the pathway and the ketoacid as a key intermediate, to explain the previous mutant/gene silencing results (see also response to reviewer #1, point 1).

We have also modified the Introduction passage detailing previous knowledge/hypotheses for the DMP enzyme, to emphasize that the sequence similarity between the wheat DMP and *Curcuma longa* CURS proteins had been noticed prior to our work.

2. Page 10: The C-terminal HDEL enables proteins to be localized in the endoplasmic reticulum (ER). Why use the marker HDEL-RFP, not the RFP-HDEL? In addition, please analyze the localization signal peptide of DMP. Pro35S: GFP-HvDMP was lost.

Both HDEL-RFP and RFP-HDEL have been used as ER marker, to our knowledge with equal success. Based on amino acid sequence prediction, HvDMP does not contain a strong signal peptide.

We tried to co-express GFP-HvDMP and ER marker HDEL-RFP in barley protoplasts, but could not find cells harboring both these constructs. This was likely due to low transformation efficiencies combined with difficulties in preparing barley protoplasts. We repeated this experiment three times, but without success, before moving to tobacco transient expression to confirm the sub-cellular localization of HvDMP.

Minor comments

3. Figure 2A, please provide the complete histogram information and the standard deviation of C₃₁ diketone.

Thanks for alerting us that this figure was difficult to read. We have improved the color contrasts to better emphasize the numbers showing the exact column height and variability. We hope this will help readers understand that the y-axis is cut at 4%, thus truncating the bar for C₃₁ diketone. We chose this truncation of the axis/bar to make the other bars better visible, since the main purpose of the figure is to illustrate the amounts of the minor homologs/isomers. (Introducing a y-axis break is not an option since it would greatly deceive the eye on the relative bar heights of minor versus major components).

4. Page 17: “ μ ” in “1 μ g of resulting total RNA” need to be Italic, “N” in “an N-terminal 6xHis-tag followed” need to be Italic.

We would be happy to italicize respective letters, but to the best of our knowledge the Greek sign for “micro” and the N/C terminus designations of proteins are typically shown in normal font.

5. The formats of ref 3, 20, 33, 34, 35, 36, 37, 40 and 50 need to be checked.

Done

Reviewer #3 (Remarks to the Author):

Review of the article Biosynthesis of barley wax β -diketones: a type-III polyketide synthase condensing two fatty acyl units. Sun et al.

This is a well written, well organized, and highly informative paper. The investigation of barley β -diketones was among the first genetic models for the study of plant wax biochemical genetics, and the presence of barley's structurally unique β -diketone class of compounds has been a focus of much interest since the 1970s beginning especially with the work of Penelope von Wettstein-Knowles. Previous publications indicate that the β -diketones themselves are important in plant environmental stress adaptation (especially for plant tolerance to drought, and supra-optimal solar radiation and heat), and thereby constitute an ecologically important class of waxes, and define overall a class of wax compounds with high potential for modification toward improving crop stress tolerance.

The authors apply a straightforward scientific approach that elucidates a novel metabolic pathway being employed by barley (and likely other plants that synthesize β -diketones) for the synthesis of the β -diketones and its derivatives the hydroxy β -diketones as well as the related compounds 2-alkanol esters. A key finding is that the 3-ketoacids are a central intermediary in these pathways, and that the barley derived enzymes Diketone Metabolism Hydrolase (DMH) and Diketone Metabolism Polyketide Synthase (DMP) are together sufficient to generate the β -diketones from plastidial acyl precursors. The authors have a strong publication record involving the chemical analysis of plant waxes, as well as related metabolic engineering studies, and the methods used in this paper are sound.

The authors provide strong evidence that the biosynthesis of β -diketones and hydroxy β -diketones are produced in a way that it does not utilize the previously described sequential 2-carbon-based elongation steps catalyzed by the Fatty Acid Elongase (FAE) pathway, and that their synthesis does not require the endoplasmic reticulum (ER)-localized β -ketoacyl-CoA synthase (KCS6). The authors demonstrated the *emr1* mutant deficient in KCS6 was able to produce β -diketones with normal chain length, even though other wax constituents on *emr1* were much shorter, and so β -diketone synthesis in barley does not require that same elongation pathway. The authors present evidence that there is not another β -diketone parallel elongation pathway, citing a previous article showing that fatty acyl chain elongation inhibitors do not inhibit elongation of β -diketones and hydroxy β -diketones, indicating that β -diketone synthesis does not involve the previously describes ER based FAE. Further, carbon isotope studies by the

authors of this paper using GC coupled with isotope-ratio mass spectrometry revealed that the β -diketones on barley are composed of all plastid generated carbons (enriched in ^{13}C), and not the ER associated carbons wherein only about 50% of the carbons would be plastidial if the chains were further elongated using the FAE pathway mechanism (wherein ^{13}C composition would be low). The authors tested these ^{13}C ratios against other wax classes in barley, providing strong evidence to support a novel synthesis that does not involve ER elongation by FAE.

Instead, the authors provide strong evidence for an alternative and novel very long chain β -diketone synthesis pathway that involves Diketone Metabolism Hydrolase (DMH) that has a thioesterase function and generates fatty acyl derivatives with functional groups on the 3rd carbon, and a Diketone Metabolism Polyketide Synthase (DMP) that performs a decarboxylative condensation of 3-ketoacid acyl chains and fatty acyl-CoA to make the β -diketones. *E. coli* and yeast expression studies were used to demonstrate both HvDMH and HvDMP preferred substrate classes and chain length specificity. It's notable that the yeast system expressing HvDMH could not make the 3-ketoacids substrate for use by HvDMP in synthesis of β -diketones, and so that is why exogenous 3-ketoacids needed to be added to test HvDMP activity. As such, this study does not directly show that HvDMH catalyzes the thioesterase reaction leading to 3-ketoacyls as would be necessary in the plant system for β -diketone production. The authors speculate that HvDMH might not interact properly (as it would in planta) with the yeast FAS complexes, or that it requires specific ACP isoforms and discriminates against yeast ACPs, which are reasonable speculations by these authors. However, HvDMH expressed in *E. coli* did hydrolyze substrates to make acyl products having 3-carbon functional groups, which provided evidence that HvDMH likely generates the 3-ketoacyls in planta. It is known that *E. coli* has very low relative amounts of saturated acyl substrates overall (relative to unsaturated), which may explain why detectable amounts of the specific 3-ketoacyls were not generated by HvDMH in the *E. coli* system.

Studies in the yeast system using deuterated C16 acids showed that HvDMP could condense (with decarboxylation) a deuterated C16 fatty acid with 3-ketoacids to make β -diketones, and so this then shows that HvDMP has the capacity to combine two long chain substrates together. HvDMP's ability to condense fatty acyl-CoAs with ketoacid substrates was further confirmed in yeast experiments using uncommon odd chain length substrates. Further in vitro/yeast studies tested chain length specificity of HvDMP to find that the HvDMP enzyme prefers the C14 3-ketoacid, and either the C14 or C16 fatty acyl-CoA. But notably HvDMP does in fact readily accept the C16 3-ketoacid. The fact the C16 3-ketoacids and C16 fatty acyl-CoAs are by far the dominant substrate pool available in planta explains why the C31 14,16-diketones are the dominant homologues found on the surface of wild-type barley sheaths, being 96% of the waxes. The functions described here for HvDMP are thus consistent with the β -diketone wax profiles observed on barley.

Finally, the authors localized both HvDMH and HvDMP using confocal microscopy, barley protoplast, and GFP constructs, showing the HvDMH enzyme was localized in the chloroplasts and HvDMP in ER membranes. They proposed a quite reasonable model for substrate transfer (and related reaction mechanisms) in the biosynthetic pathway.

They also describe for the first time unbranched even chain length β -diketones, likely derived from an initial C3 rather than the more common C2 moiety. They then noted that the locations of the functional

groups on these even chain β -diketones are consistent with a main finding of this report that the C16 3-ketoacids are a major intermediate in this β -diketone pathway.

Although not necessary for publication in the opinion of this reviewer (as will be described below), additional evidence that HvDMH directly catalyzed synthesis of the 3-ketoacyls in yeast or *E. coli* (or another system) would have helped strengthen the narrative that two enzymes are sufficient for β -diketone synthesis. Notwithstanding, that the *E. coli* studies showed that HvDMH produced closely related 3-carbon modified acyl derivatives does provide good evidence that the HvDMH does in fact generate 3-ketoacyls in plants. Regardless, the presentation of only indirect evidence of HvDMH function in 3-ketoacyl synthesis does not detract from the overall main discoveries of this paper, which is the determination that the HvDMP enzyme generates the β -diketones, and that this is done using a novel reaction that does not involve the well-known sequential C2 FAE elongation steps, but instead uses HvDMP to perform a decarboxylative condensation reaction of two long chain plastidial derived substrates. There are no previous reports of such a condensation reaction of long chain acyl compounds in plants, and this is the first report of the synthesis of very long chain aliphatic compounds that does not involve C2 elongation steps.

We thank the reviewer for their very detailed comments! The remark above that our study “does not directly show that HvDMH catalyzes the thioesterase reaction” as well as the comments here on additional evidence prompted us to perform further *in vitro* experiments to assay the HvDMH activity (see also response to reviewer #1, point 2). A new figure (fig 4) shows respective results, and an additional paragraph in the Results section on HvDMH explains the findings.

The findings here represent novel and remarkable discoveries on their own and are clearly supported by the data and its interpretation.

Some minor edits requested:

Under carbon isotope analysis in methods, lines 2 and 3, recommend simply rewording as “were stained with”.

Done

Remove the word “at” in line 15 for Fig 1 caption.

Done

In Fig 2A, the green colors in the legend are a bit difficult to align to the bars. Making the colors more different will improve the ease of interpretation. The colors in Fig 2 C are also difficult to visualize. Point symbols should be enlarged

Thanks for alerting us that both these figure panels were difficult to read. We have changed the colors and symbol sizes to help readers distinguish and properly assign data points according to the legend.

REVIEWERS' COMMENTS

Reviewer #1 (Remarks to the Author):

This reviewer is satisfied with the additional experiments and revised manuscript. This manuscript is now suitable for publication in Nature Communications.

Reviewer #2 (Remarks to the Author):

The authors addressed the reviewers' concerns. Publish as it is.

Reviewer #3 (Remarks to the Author):

This reviewer was concerned that the study did not directly show that HvDMH catalyzes the thioesterase reaction leading to 3-ketoacyls as would be necessary in the plant system for β -diketone production. Nevertheless, this reviewer felt that the original submission contained strong evidence that HvDMH did in fact have such activity as claimed. To strengthen the authors conclusions about HvDMH activity, the authors performed additional in vitro experiments to assess and confirm HvDMH function. The authors have adequately addressed this concern.